# Translate Policy to Language: Flow Matching Generated Rewards for LLM Explanations

**Xinyi Yang**[1,*] **Liang Zeng**[2] , **Heng Dong**[2], **Chao Yu**[3], **Xiaoran Wu**[4], **Huazhong Yang**[3], **Yu Wang**[3], **Milind Tambe**[5], **Tonghan Wang**[6,†]

[1] AIPD, Tencent, Shenzhen, China
[2] IIIS, Tsinghua University, Beijing, China
[3] EE, Tsinghua University, Beijing, China
[4] CS, Tsinghua University, Beijing, China
[5] SEAS, Harvard University, Cambridge, USA
[6] College of AI, Tsinghua University, Beijing, China

## Abstract

As humans increasingly share environments with diverse agents powered by RL, LLMs, and beyond, the ability to explain agent policies in natural language is vital for reliable coexistence. We introduce a general-purpose framework that trains explanation-generating LLMs via reinforcement learning from AI feedback, with distributional rewards generated by generative continuous normalizing flows (CNFs). CNFs capture the pluralistic and probabilistic nature of human judgments about explanations. Moreover, under mild assumptions, CNFs provably bound deviations from true human reward distributions when trained on noisy proxy rewards from LLMs. We design a specialized CNF architecture that selectively attends to linguistic cues in the decision context and explanations when generating rewards. Human and LLM evaluators find that our method delivers explanations that enable more accurate predictions of true agent decisions, exhibit greater logical soundness and actionability, and impose lower cognitive load than explanations trained with proxy LLM rewards or state-of-the-art RLHF and RLAIF baselines.

## 1 Introduction

Intelligent agents, ranging from reinforcement learning (RL) agents (Kiran et al., 2021; Zhao et al., 2021; Liu & Zhu, 2024; Qiu et al., 2024), large language models (LLMs) (Yao et al., 2022; Shinn et al., 2024; Wang et al., 2023a; Kang et al., 2020), to robotic systems (Ismail et al., 2018), are becoming increasingly intertwined with daily lives (Wang et al., 2024a). To foster transparency, safety, and alignment, a promising avenue is communicating the reasoning behind actions or decisions generated by agent policies in natural language (Lazaridou et al., 2016). A good explanation provides a rationale that helps recover the underlying policy, enabling third parties to infer the decision. Such explanations allow humans to comprehend decisions, provide meaningful feedback, and intervene when necessary, ultimately enhancing trust and reliability within intelligent ecosystems (Cambria et al., 2023).

This effort is situated within the broader field of explainable AI and RL (Arrieta et al., 2020; Carvalho et al., 2019; Ehsan et al., 2019; Gunning, 2017; Ras et al., 2018; Gilpin et al., 2018), which encompasses methods such as causal inference and contrastive learning. Nevertheless, existing natural language explanation methods are often tailored to specific application scenarios, including self-driving (Cai et al., 2024), recommender systems (Lubos et al., 2024), stock prediction (Koa et al., 2024), robotics (Lu et al., 2023), and autonomous navigation (Trigg et al., 2024). To circumvent the extensive task- and agent-specific engineering, there is a need for a generalizable and task-agnostic explanation method. We achieve this by concealing agent decisions from an EXPLANATION LLM, prompting it solely with decision contexts, and training it to recover a specific underlying policy and generate

---

*The work was done while the author was a PhD student at Tsinghua.
†Correspondence to tonghanwang1996@gmail.com

explanations that reflect that policy's behavior. This approach forces the LLM to reason only from contextual information, independent of task or agent specifics, and thereby achieves generalizability.

In reinforcement learning from human feedback (RLHF), such third-party evaluations are provided by humans. In explanation tasks, human feedback is inherently pluralistic (Feng et al., 2024; Sorensen et al., 2024a; Kirk et al., 2024; Sorensen et al., 2024b) and probabilistic (Lou et al., 2024; Dorka, 2024), as explanations are open-ended and subject to wide variation in individual judgments. This pluralism necessitates the costly collection of diverse human feedback in RLHF. Reinforcement learning from AI feedback (RLAIF) (Bai et al., 2022b; Pace et al., 2024; Liu et al., 2024b) economizes human feedback, typically by adapting auxiliary LLMs to generate *proxy rewards* that approximate human feedback. Although early work (Li et al., 2024a) shows promise, current RLAIF approaches have not yet been rigorously studied on how to generate distributional rewards that explicitly manage proxy errors relative to human reward distributions.

We close this gap by extending RLAIF with generative continuous normalizing flows (CNFs) (Chen et al., 2018; Lipman et al., 2023) as reward models. CNFs map base noise distributions to complex distributions, a process we exploit to represent reward distributions for training the EXPLANATION LLM. We train a CNF to fit reward samples generated by separate PROXY LLMs. This method enjoys the theoretical benefit that, assuming the noise in these rewards generated by PROXY LLMs has the same functional form as the CNF's base distribution (e.g., Gaussian), the CNF provably bounds its deviation from the underlying human reward distribution. Intuitively, the PROXY LLM acts as a forward process that injects noise into human rewards, and the CNF learns its inverse, partially denoising and recovering the underlying human rewards. These analyses extend to all generative models based on ordinary differential equations (ODEs), including score-based diffusion models (Song et al., 2021). Among them, we choose rectified flow (RF) (Liu et al., 2022; Albergo & Vanden-Eijnden, 2023) for its fast inference and high distributional fidelity. Appx. D demonstrates the proof ideas of this theoretical benefit.

To operationalize these theoretical benefits, we condition the RF reward model on decision contexts and explanations. We learn to transform inputs of the RF model into *flow tokens* that cross-attend to latent embeddings of decision contexts and explanations. The resulting attention summary informs the flow's output, allowing it to selectively integrate linguistic and contextual cues when assigning reward distributions.

We evaluate our method on both RL tasks (`SMAC`, (Samvelyan et al., 2019)) and LLM tasks (`MMLU` (Hendrycks et al., 2020), `MathQA` (Amini et al., 2019)). Removing the rectified flow and directly using rewards from proxy LLMs decreases performance by 6.9%-12.4%, demonstrating the effectiveness of the generated rewards. Moreover, our method outperforms `SFT`, `RLHF` (Bai et al., 2022a), `RLAIF` (Bai et al., 2022b), and reasoning baselines (Trung et al., 2024) by 1.9%-18.2% across all tasks and backbone models. Human evaluations further confirm the high quality of our generated explanations, which receive the highest scores of accuracy, logicality, actionability, and cognitive load.

## 1.1 RELATED WORKS

**LLM explanations**. Methods such as AMPLIFY(Krishna et al., 2024), Self-Explain(Rajagopal et al., 2021), and Summarize and Score (SASC) (Singh et al., 2023) generate concise rationales based on agent decisions, sometimes accompanied by an explanation score to assess reliability. Unlike these methods, our approach trains LLMs to generate explanations without knowing agent decisions a priori. Chain-of-Thought (CoT) prompting (Wei et al., 2022) is a widely adopted in-context learning technique that relies on step-by-step explanations or reasoning to enhance decision-making. Self-Taught Reasoner (STaR) (Zelikman et al., 2022) introduces an iterative refinement method, where a model improves its own explanations through self-generated rationales. While these methods are prompting-based and do not require additional training, optimization-based CoT methods like `CoT+SFT` (Wei et al., 2022) and `ReFT` (Trung et al., 2024) have been developed. We compare against these optimization-based methods in our experiments for their proven powerful reasoning capability.

**RLHF**. Reinforcement learning from human feedback (`RLHF`) (Bai et al., 2022a; Wang et al., 2023b; Ouyang et al., 2022; Dong et al., 2024) is critical to aligning LLM behavior with human preferences such as helpfulness, harmlessness, and honesty (Ganguli et al., 2022; Achiam et al., 2023; Team et al., 2023). An RL-based method uses a reward model (Liu et al., 2024b) to approximate human preferences. Given a dataset of preferred versus rejected responses, a reward model can be trained

using the standard Bradley-Terry model (Bradley & Terry, 1952) with a pairwise ranking loss. The policy model (LLM) can then be fine-tuned by learned rewards via PPO (Schulman et al., 2017). However, training a reward model can be costly. Direct preference learning (DPO) (Rafailov et al., 2024) enables direct training on preference data, which can be adapted to different human utility models (KTO, (Ethayarajh et al., 2024)). We compare against PPO, DPO, and KTO in our experiments.

**RLAIF**. (Bai et al., 2022b) introduces RLAIF, integrating LLM-generated preferences with human-defined principles for improved reward modeling. Subsequent studies, such as Skywork (Liu et al., 2024b) and West-of-N (Pace et al., 2024), further enhance this area by refining methods for curating preference data. While these approaches still partially rely on human annotations, other methods entirely exclude human data (Wang et al., 2024b; Wu et al., 2024b). In our experiments, we include comparisons with Skywork (Liu et al., 2024b), the leading RLAIF algorithm on the RewardBench leaderboard (Lambert et al., 2024).

Several recent RLHF and RLAIF studies train *distributional reward models* for LLM post-training. QRM (Dorka, 2024) employs quantile regression to estimate multimodal preference distributions. DPRM (Li et al., 2024a) represents multiple preferences by a categorical distribution and approximates the target distribution via an optimal transport objective. In our setting, both methods would need to discretize explanation feedback–the probability that a third party can infer the true agent decision from explanations–and then fit a distribution on that discretized space. By contrast, our method seeks to model continuous feedback, learning distributions without discretization. URM (Lou et al., 2024) parameterizes Gaussian distributions to model disentangled human preference attributes, whereas our approach accommodates arbitrary reward distributions.

Please see Appx. A for more related works on **Diffusion in Transformer**, **Cross-attention**, **Explainable AI/RL**, and **LLM Training in Explanations**.

## 2 PRELIMINARY

**Rectified Flow** (Liu et al., 2022; Albergo & Vanden-Eijnden, 2023) emerges as a powerful generative model and has recently served as the basis for popular commercial tools like Stable Diffusion 3 (Stability AI, 2023). Based on continuous normalizing flow (CNF) (Chen et al., 2018; Lipman et al., 2023), it models a generative process as an Ordinary Differential Equation (ODE). Formally, a CNF transports an input $\boldsymbol{z}_0 \in \mathbb{R}^d$ to $\boldsymbol{z}_t = \phi(t, \boldsymbol{z}_0)$ at time $t \in [0, 1]$ via the ODE:

$$\frac{d}{dt}\phi(t, \boldsymbol{z}_0) = \varphi\left(t, \phi(t, \boldsymbol{z}_0)\right). \tag{1}$$

$\phi : [0, 1] \times \mathbb{R}^d \to \mathbb{R}^d$ is the *flow*, and the *vector field* $\varphi : [0, 1] \times \mathbb{R}^d \to \mathbb{R}^d$ specifies the change rate of the state $\boldsymbol{z}_t$. (Chen et al., 2018) suggests representing the vector field $\varphi$ with a neural network. The flow $\phi$ transforms an initial random variable $Z_0 \sim p_0(\boldsymbol{z}_0)$ to $Z_1 \sim p_1(\boldsymbol{z}_1)$ at time $t = 1$. Rectified flow drives $\phi$ to follow the linear path in the direction $(Z_1 - Z_0)$ as much as possible by optimizing the objective: $\min_\varphi \int_0^1 \mathbb{E}\left[\|(Z_1 - Z_0) - \varphi(t, Z_t)\|^2\right] dt$, where $Z_t = t \cdot Z_1 + (1 - t) \cdot Z_0$ is the linear interpolation of $Z_0$ and $Z_1$. Typically, the vector field network $\varphi$ is implemented as a U-Net (Ronneberger et al., 2015) for image inputs or an MLP for vector inputs (Wang et al., 2024c).

**Transformer**. The Transformer architecture (Vaswani et al., 2017) is foundational to recent progress in LLMs (Liu et al., 2024a; Zeng et al., 2024; Yang et al., 2024a; Team et al., 2023). For an input token sequence $x = (\boldsymbol{x}^{(1)}, \dots, \boldsymbol{x}^{(N)})$, let $E^{(n)} = [e(\boldsymbol{x}^{(1)}), \dots, e(\boldsymbol{x}^{(n)})]$ denote the sequence of token embeddings up to position $n$, where $e(\cdot)$ is the token embedding function. A standard LLM generates its output by

$$\boldsymbol{H}^{(n)} = \text{TRANSFORMER}\left(E^{(n)}\right), \quad M\left(\boldsymbol{y}^{(n+1)} \mid \boldsymbol{x}^{(\leq n)}\right) = \boldsymbol{W}\boldsymbol{h}^{(n)}, \tag{2}$$

where $\boldsymbol{H}^{(n)} \in \mathbb{R}^{n \times d}$ denotes the last layer hidden states for the first $n$ tokens, with $d$ representing the hidden dimensionality. The last layer hidden state at position $n$ is $\boldsymbol{h}^{(n)} = \boldsymbol{H}^{(n)}[n, :]$. We denote the output projection matrix by $\boldsymbol{W}$, the generation logits by $M$, and the output tokens by $\boldsymbol{y}$.

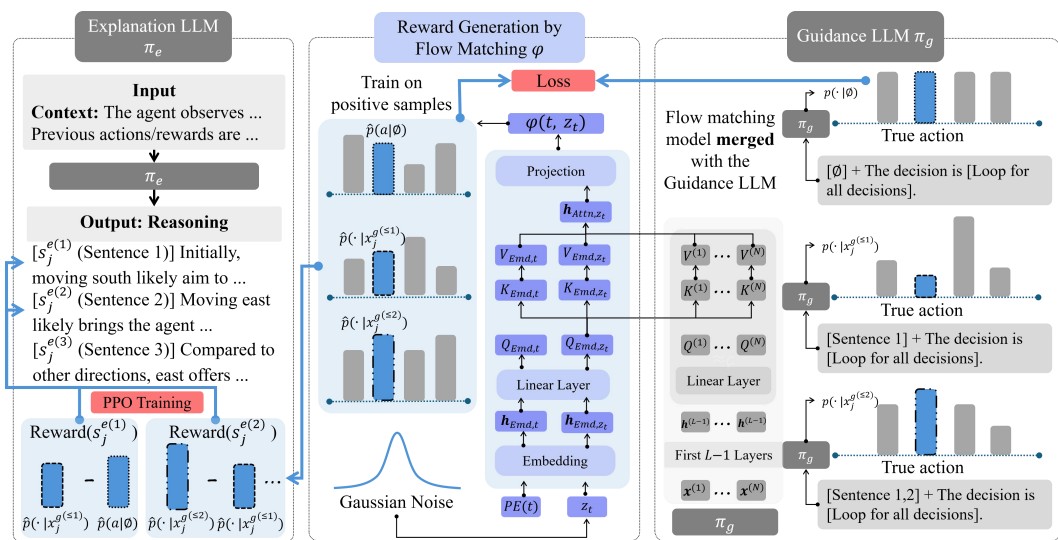

Figure 1: An overview of our method. (Left) We prompt an EXPLANATION LLM to generate reasoning about an agent decision based on the context information. Our focus is on whether a third party can infer the actual decision from this explanation. (Middle) We employ a rectified flow model $\varphi$ to generate rewards capturing how likely the actual decision appears as a plausible outcome after each sentence of the explanation. Per-sentence rewards for training the EXPLANATION LLM are the changes in the probability of the actual decision (highlighted in blue). (Right) The architecture and training of the rectified flow $\varphi$ are based on a PROXY LLM. The PROXY LLM provides positive samples, where, with the context and explanation as input, it can produce a distribution $p$ that assigns the highest probability to the actual decision. The rectified flow $\varphi$ is trained to produce such distributions $p$, with a cross-attention layer in its middle that selectively leverages information from the PROXY LLM input, enabling generalization to negative samples.

## 3 METHOD

Our goal is to train an EXPLANATION LLM $\pi_e(\theta_e)$ to recover a specific underlying policy and generate natural language explanations that reflect that policy's decision of RL agents and LLMs given only context. In RL, the decision is an RL action, and the context is the trajectory of preceding states, actions, and rewards. In LLM tasks, the decision could be, for example, the selected option for a multiple-choice question, and the context is the question itself. Given a set $\mathcal{D}_e = \{(a_j, c_j)\}_{j=1}^{J}$ of decisions $a_j$ and their context $c_j$, we construct a prompt $x_j^e$ for $\pi_e(\theta_e)$: Given $\left[\texttt{Context } \texttt{c\_j}\right]$. Please analyze the reasoning behind the agent decision based on the context. The EXPLANATION LLM then generates the corresponding explanation $y_j^e = \pi_e(x_j^e; \theta_e)$.

To prevent superficial explanations that simply restate the decision, we intentionally exclude the actual decision $a_j$ from the prompt $x_j^e$. For the generated content to be effective explanations, we train the EXPLANATION LLM using RL with rewards generated by a rectified flow $\varphi(\theta_\varphi)$. These rewards are probabilistic, reflecting the fact that human individuals have different opinions about the efficacy of the same explanation. Different explanation models correspond to different policies differentiated by training with distinct rewards. Fig. 1 illustrates our learning framework. The architecture and training of $\varphi(\theta_\varphi)$ are described in the following subsection.

### 3.1 RECTIFIED FLOW REWARD MODEL

We aim at generating rewards that reflect how likely human individuals can infer correct decisions from explanations. To save costly human feedback (Bai et al., 2022b), we use LLMs as proxy feedback providers when training the rectified flow reward model. Proxy LLM feedback may deviate from real human feedback. We empirically show the influence of these deviations as a baseline method Proxy LLM in Tab. 2 and formally analyze how the rectified flow manages these deviations in Sec. 4.

Specifically, we query a PROXY LLM with the prompt $x_j^g$: `Given [Context c_j], the reasoning is [y_j^e]. Thus, the decision is [a decision a]`. For each candidate decision $a$, we measure the likelihood of it being a plausible outcome based on the token-level logits $M(a|x_j^g)$ (Eq. 2), which indicate how likely the PROXY LLM is to output $a$ at the end of $x_j^g$. If $a$ spans multiple tokens, we compute the mean of their logits (Yang et al., 2022).

To capture the probabilistic nature of judgments on an explanation, we issue the same query for $K$ independent PROXY LLMs, collecting logits $\{M_{j,k}(\cdot|x_j^g)\}_{k=1}^K$. The rectified flow $\varphi(\theta_\varphi)$ is trained by minimizing the following loss function:

$$\mathcal{L}_{\text{FLOW}}(\theta_\varphi) = \mathbb{E}_{\boldsymbol{z}_0 \sim Z_0, (c_j, y_j^e, M_{j,k}) \sim \mathcal{D}_p, \boldsymbol{z}_1 = M_{j,k}, t \sim [0,1]} \left[ \|(\boldsymbol{z}_1 - \boldsymbol{z}_0) - \varphi(t, \boldsymbol{z}_t | c_j, y_j^e; \theta_\varphi)\|^2 \right]. \quad (3)$$

where $\mathcal{D}_p = \{(c_j, y_j^e, \{M_{j,k}\}_{k=1}^K)\}_{j=1}^J$ is the training dataset and $\boldsymbol{z}_t = t \cdot \boldsymbol{z}_1 + (1-t) \cdot \boldsymbol{z}_0$, for $t \in [0,1]$. The flow model transports each initial random sample $\boldsymbol{z}_0 \in \mathbb{R}^{|\mathcal{A}|}$ drawn from $Z_0$ to a logit vector of the same dimension, where $\mathcal{A}$ is the set of all candidate decisions. With initial samples from $Z_0$, $\varphi(\theta_\varphi)$ effectively constructs a distribution over logits. We define rewards as the logits corresponding to the ground-truth decision. Selecting them from the generated logits yields a corresponding distribution over rewards. We discuss how to use this reward distribution in Sec. 3.2.

The choice of $Z_0$ is critical for the rectified flow to handle noise in the PROXY LLM, as we will discuss in Sec. 4.

**Rectified Flow Model Architecture**. As in a standard rectified flow model, the input to $\varphi$ includes (1) the current state $\boldsymbol{z}_t \in \mathbb{R}^{|\mathcal{A}|}$, $t \in [0,1]$, with $\boldsymbol{z}_0$ sampled from $Z_0$; and (2) a positional encoding $PE(t)$ corresponding to the ODE time $t \in [0,1]$. Apart from these, the inputs also include (3) context $c_j$ and explanations $y_j^e$. This requires that $\varphi$ can understand the linguistic cues in $c_j$ and $y_j^e$, which is beyond the capacity of typical rectified flow models based on fully-connected networks (Wang et al., 2024c) or U-Nets (Liu et al., 2022). To solve this problem, we propose to embed the rectified flow model $\varphi$ into an LLM.

*Embedding*. The context $c_j$ and explanations $y_j^e$ are encoded by the EXPLANATION LLM, and we use its last layer latent states. ReLU-activated, layer-normalized MLPs $\varphi_{\text{EMB}}$ are used to project $\boldsymbol{z}_t$ and $PE(t)$ to have the same dimension as the LLM tokens. $PE(t)$ and $\boldsymbol{z}_t$ use two separate embedding MLPs: $\boldsymbol{h}_{\text{EMB},t} = \varphi_{\text{EMB},t}(PE(t)), \boldsymbol{h}_{\text{EMB},\boldsymbol{z}_t} = \varphi_{\text{EMB},\boldsymbol{z}_t}(\boldsymbol{z}_t)$. The resulting embeddings, stacked as $\boldsymbol{H}_{\text{EMB}} = (\boldsymbol{h}_{\text{EMB},t}, \boldsymbol{h}_{\text{EMB},\boldsymbol{z}_t})^\top \in \mathbb{R}^{2 \times d}$, are called *flow tokens*.

*Cross-Attention*. We use the LLM's last layer weight matrices $(\boldsymbol{W}_Q, \boldsymbol{W}_K, \boldsymbol{W}_V)$ to generate queries, keys, and values of the flow tokens: $(\boldsymbol{Q}_{\text{EMB}}, \boldsymbol{K}_{\text{EMB}}, \boldsymbol{V}_{\text{EMB}}) = \boldsymbol{H}_{\text{EMB}}(\boldsymbol{W}_Q, \boldsymbol{W}_K, \boldsymbol{W}_V)$, which are concatenated with the last layer hidden states of the $c_j$ and $y_j^e$ to calculate cross-attention. More details are discussed in Appx. C.

*Projector*. Define $\boldsymbol{h}_{\text{ATTN},\boldsymbol{z}_t}$ as the latent state of the flow token $\boldsymbol{h}_{\text{EMB},\boldsymbol{z}_t}$ after cross-attention. This state has incorporated the ODE time $t$ and explanatory information through cross-attention. Progressing from this state, we use a fully-connected network $\varphi_{\text{PROJ}} : \mathbb{R}^d \to \mathbb{R}^{|\mathcal{A}|}$ with ReLU activation and layer normalization to generate the vector field $\varphi = \varphi_{\text{PROJ}}(\boldsymbol{h}_{\text{ATTN},\boldsymbol{z}_t})$. We find that skip-layer connections are important for training stability, and we append the inputs $\boldsymbol{z}_t$ and $t$ to the hidden layers of $\varphi_{\text{PROJ}}$.

## 3.2 OVERALL TRAINING PROCEDURE

Once the rectified flow model $\varphi$ is trained, we use it to generate rewards for samples in $\mathcal{D}_e$. Concretely, we solve the ODE $d\boldsymbol{z}_t = \varphi(t, \boldsymbol{z}_t)dt$ starting from $\boldsymbol{z}_0$ sampled from the initial distribution. An advantage of rectified flow is that the vector field $\varphi$ is encouraged to be straight lines, allowing efficient and accurate solution of the ODE in a few steps. The solution $\boldsymbol{z}_1$ (at time 1) is taken as the estimated logit $M_{\text{FLOW}}(\cdot|x_j^g) = \boldsymbol{z}_1 = \boldsymbol{z}_0 + \int_0^1 \varphi(t, \boldsymbol{z}_t)dt$.

Given different initial samples $\boldsymbol{z}_0$, we might get different logits $M_{\text{FLOW}}$, which reflect probabilistic opinions. We use the average logit of the true decision $M_{\text{FLOW}}(a_j|x_j^g)$ as the reward. We then use PPO to train the EXPLANATION LLM to improve its explanations that achieve higher rewards from the rectified flow model. The rectified flow model and EXPLANATION LLM are trained alternately until convergence. The EXPLANATION LLM is fine-tuned using PPO with LoRA. The rectified flow model consists of a frozen LLM backbone and two trainable MLPs: $\varphi_{\text{EMB}}$ and $\varphi_{\text{PROJ}}$.

**Implementation details**. In practice, we adopt rejection sampling (Touvron et al., 2023) when training the flow model. Specifically, we only use samples where PROXY LLMs assign the highest likelihood to the true decision: $a_j = \arg\max_a M(a|x_j^{\bar{g}})$. To mitigate the influence of large-magnitude logit values, we apply a softmax activation to $M(\cdot|x_j^g)$ before using it in the loss function (Eq. 3). Moreover, our model supports dense rewards. The logits change as we feed sentences in an explanation incrementally. We define per-sentence rewards as the changes of the true decision $a_j$'s logit after each newly added sentence. The sentence-level reward is used in addition to the sparse reward, where a reward of 1 is assigned only to the last token of the explanation if the predicted decision is correct; otherwise, the reward is 0. More details are discussed in Appx. C. We study the effects of rejection sampling and sentence-level rewards in Sec. 5.

## 4 THEORETICAL ANALYSIS

In this section, we analyze the usage of a rectified flow model to approximate rewards.

Let $p(\cdot|y_j^e)$ be the unknown human reward distribution for the explanation $y_j^e$. As discussed, using LLMs as proxy human reward approximators might introduce errors. We first assume a Gaussian additive error model, and $p_0 = \mathcal{N}(0, \sigma_r^2 \boldsymbol{I}_{|\mathcal{A}|})$ is the corruption noise. The LLM-generated distribution is given by the convolution $\hat{p} = p * p_0$. For rectified flow, we adopt an initial distribution of the same functional form as the corruption noise: $q_0 = \mathcal{N}(0, \sigma^2 \boldsymbol{I}_{|\mathcal{A}|})$. Define the terminal distribution of the flow $\phi_t$ as $q_1 = \phi_{1\sharp} q_0$. We can prove the following theorem.

**Theorem 1.** *[Error bound for recovering the human reward distribution with a rectified flow] The flow is trained so that the fitting error to the proxy LLM reward distribution, measured in quadratic Wasserstein distance, satisfies $W_2(\hat{p}, q_1) \leq \varepsilon$. $p_{\text{FLOW}} := \phi_{1\sharp} p_0$ is the denoised candidate obtained by pushing $p_0$ through the learned flow. Errors relative to the true human distribution are bounded by*

$$W_2(p_{\text{FLOW}}, p) \leq \varepsilon + L\sqrt{|\mathcal{A}|}\left|\sigma - \sigma_r\right|,$$

*where $L = \exp\left(\int_0^1 \|\nabla\varphi_t\|_\infty \, dt\right)$ is the Lipschitz constant. In many architectures, $L \approx 1$.*

The proof in Appx. D applies to any ODE-based generative models like score-based diffusion models. For other learning methods, such as previously proposed distributional reward modeling approaches (Dorka, 2024; Lou et al., 2024; Li et al., 2024a), we have the following lemma applicable when they learn from AI feedback.

**Lemma 2.** *[Error bound without a rectified flow] Without a rectified flow model, the error of a learning model approximating $\hat{p}$ relative to the true human distribution is bounded by*

$$W_2(\tilde{p}, p) \leq \varepsilon' + \sqrt{|\mathcal{A}|}\left|\sigma_r\right|,$$

*where $\varepsilon'$ is the model's approximation error with respect to $\hat{p}$. Moreover, the inherent error in the proxy LLM reward distribution is $W_2(\hat{p}, p) = \sqrt{|\mathcal{A}|}\left|\sigma_r\right|$.*

The major takeaway is that a rectified flow turns an unavoidable bias term $W_2(\hat{p}, p) = \sqrt{|\mathcal{A}|}\left|\sigma_r\right|$ into a controllable one as long as the initial distribution of the flow and the error model share the same functional form. A reasonable estimation of hyperparameters, e.g., ($\sigma \approx \sigma_r$), guarantees that a rectified model can well recover the true human reward distribution. We further provide a theoretical analysis in Appx. D for the case where $q_0$ and $p_0$ follow different functional forms. Figure 3 in Appx. D illustrates the proof ideas of our theorems.

## 5 EXPERIMENT

**Benchmark**. We evaluate our method on three testbeds: (1) SMAC (StarCraft Multiagent Challenge (Samvelyan et al., 2019)) is an RL benchmark based on the real-time multi-agent strategy game StarCraft II that emphasizes micromanagement challenges. Explaining a decision here requires analyzing the status of ally and enemy units, such as their remaining health points and positions, and inferring their intentions from trajectories. (2) AI2-THOR (Kolve et al., 2017) is a widely used embodied AI benchmark in RL research (Li et al., 2024b; Wu et al., 2021), featuring household activity scenarios that demand multi-step planning, multi-robot coordination, and interaction with

dynamic environments. We use the *Complex Tasks* split from SMART-LLM (Kannan et al., 2024) and expand it from 8 to 59 tasks. (3) `MMLU` (Massive Multitask Language Understanding (Hendrycks et al., 2020)) is a multiple-choice QA benchmark for LLMs. We choose 4 challenging Professional Knowledge subsets (Professional Medicine (272 samples), Professional Law (1.53K samples), Professional Accounting (282 samples), and Professional Psychology (612 samples)). (4) `MathQA` (Amini et al., 2019) is a large-scale multiple-choice QA benchmark for math word problems, comprising 30K training samples and 3K testing samples. Please refer to Appx. F for more details.

**Baselines**. We evaluate our method against seven baseline families. **(1) Proxy LLM**. This baseline directly uses rewards generated by the PROXY LLM to train the EXPLANATION LLM. **(2) SFT** is a popular paradigm for adapting LLMs to downstream tasks. **(3) RLHF**. We consider `PPO` (Xu et al., 2024), `DPO` (Rafailov et al., 2024) and `KTO` (Ethayarajh et al., 2024), using the TRL implementations (von Werra et al., 2020). **(4) Reasoning Frameworks**. We compare against `CoT+SFT` (Wei et al., 2022) and `ReFT` (Trung et al., 2024). **(5) RLAIF**. We report results for SKYWORK (Skywork-Reward-Llama-3.1-8B-v0.2) (Liu et al., 2024b), which ranks in the top two 8B models on the RewardBench leaderboard (Lambert et al., 2024). **(6) DeepSeek**. We compare with DeepSeek-R1-Distill-Llama-8B (Guo et al., 2025). **(7) o3-mini**. We directly use `o3-mini` (OpenAI, 2025) for evaluation without fine-tuning.

**Ablations**. **(1)** `Ours w/o Attn`. We remove the cross-attention layer in the reward model $\varphi$ and use a fully-connected network to generate the vector field. **(2)** `Ours` (`Sparse Reward`). We use a single reward for the last token of the explanation instead of sentence-level rewards. **(3)** `PPO` (`Sentence Reward`). We apply sentence-level rewards to `PPO`. **(4)** `Ours w/o Flow` is the same as the baseline `Proxy LLM`.

**Models and Training**. By default, we use Llama-3.1-8B-Instruct (Dubey et al., 2024) as the backbone of the EXPLANATION LLM. We also test different backbone models, including Llama-3.1-8B, Qwen2.5-7B-Instruct (Yang et al., 2024a) and Gemma-2-2B-It (Lieberum et al., 2024) to assess the robustness of our method.

All methods except `o3-mini` are trained on a shared synthetic dataset generated by o1-mini (Jaech et al., 2024) and verified by human evaluators. We prompt o1-mini to generate paired *Preferred explanations* given ground-truth decisions and *Rejected explanations* given randomly selected incorrect decisions. The training dataset for `CoT+SFT` and `ReFT` is further enhanced by asking o1-mini to adopt the `CoT` paradigm. We didn't directly use established instruction-tuning datasets because they provide preference data for answers, not explanations. To evaluate the quality of our synthetic dataset, 12 participants (2 females, 10 males, aged 20–40) are invited to rate 5% randomly chosen samples from each benchmark and their corresponding explanations on a 1–4 scale based on decision inferability. As shown in Tab. 1, the results (score mean ± variance) indicate that preferred samples significantly enhance human inference accuracy compared to rejected samples, confirming the reliability of our synthetic dataset.

Table 1: Human evaluation of our constructed datasets.

| Preferred↑ | Rejected↓ |
| --- | --- |
| 3.53(0.53) | 2.26(0.76) |

All methods except `o3-mini` undergo three epochs of SFT on the preferred subset. Baselines (`PPO`, `DPO`, `KTO`, `Skywork` and `ReFT`) then perform preference optimization with the full preferred/rejected pairs. For our approach, the EXPLANATION LLM and the rectified flow model are trained iteratively for two rounds. As shown in Fig. 2(d), additional rounds provide only marginal gains. For the rectified flow model, we choose $\sigma = 1$. We use $K = 3$ independent PROXY LLMs: Llama-3.1-8B-Instruct, Qwen2.5-7B-Instruct and Gemma-2-2B-It. Complete training details and hyperparameter settings can be found in Appx. F.

**Evaluation and Metrics**. For LLM evaluation, an explanation generated by the EXPLANATION LLM is considered effective if GPT-4o correctly infers the ground-truth decision based on it, with accuracy (ACC) measured as the proportion of effective explanations. We also consider the success rate of the whole task as the evaluation metric. For human evaluation, we recruit 45 participants (30 males, 15 females, aged 20-40) from diverse backgrounds (engineering, chemistry, machine learning, and economics). Each participant reviews 3 randomly selected samples from the benchmark and predicts the ground-truth labels. We report prediction accuracy and the percentage of participants rating each method the highest logical soundness, the highest actionability, and the lowest cognitive load. More details can be found in Appx. F.

## 5.1 COMPARISONS AGAINST BASELINES

**LLM Evaluation**. Tab. 2 reports the results. For `SMAC`, `MMLU`, and `MathQA` benchmark, compared to `Proxy LLM`, our method achieves 12.4%, 6.9%, and 11.0% higher ACC on these datasets, respectively. These results support our theoretical analyses that rewards from PROXY LLMs yield ineffective optimization, while the rectified flow model better handles noisy proxy rewards. Among `RLHF` baselines, `KTO` and `DPO` perform relatively well due to strong human utility models. In Reasoning Frameworks, although `ReFT` improves ACC to 72.2%, 74.3%, and 76.3% on these datasets, it still lags behind our method. `Skywork`, `DeepSeek` and `o3-mini` also underperform our method. The performance of our method remains significantly better than baselines in the challenging `AI2-THOR` benchmark. It improves the success rate by 7.4%, 6.0%, and 2.5% over `KTO`, `ReFT` and `o3-mini`, respectively. In summary, our method consistently surpasses all baselines, demonstrating its effectiveness in generating more reasonable explanations that support inferring correct decisions.

Table 2: Accuracy results of our method and baselines on `SMAC`, `MMLU` and `MathQA`. Success rate of our method and baselines on `AI2-THOR`.

| Datasets | Ours | SFT | Proxy LLM | RLHF | | | Reasoning | | RLAIF | DeepSeek | o3-mini |
|---|---|---|---|---|---|---|---|---|---|---|---|
| | | | | PPO | DPO | KTO | CoT+SFT | ReFT | Skywork | | |
| SMAC | **0.764** | 0.582 | 0.640 | 0.653 | 0.713 | 0.721 | 0.642 | 0.722 | 0.692 | 0.658 | 0.455 |
| MMLU | **0.772** | 0.689 | 0.703 | 0.723 | 0.746 | 0.753 | 0.705 | 0.743 | 0.737 | 0.721 | 0.707 |
| MathQA | **0.804** | 0.667 | 0.694 | 0.719 | 0.753 | 0.758 | 0.712 | 0.763 | 0.729 | 0.718 | 0.739 |
| AI2-THOR | **0.702** | 0.447 | \ | \ | \ | 0.628 | \ | 0.642 | 0.483 | 0.476 | 0.677 |

**Human Evaluation**. In LLM evaluation, `DPO` and `KTO` lead the `RLHF` baselines, `ReFT` tops the reasoning frameworks. Therefore, for human evaluation, we compare our method against `DPO` and `ReFT`. Results on MathQA are shown in Tab. 3. Our method achieves the highest accuracy, exceeding the strongest baseline by 25.7%. Around 60% of participants judge our explanations to be the most logically sound and the least cognitively demanding, indicating improved decision comprehension.

Table 3: Human evaluations on `MathQA`, where Cognitive denotes cognitive load.

| Methods | ACC | Logic | Actionable | Cognitive |
|---|---|---|---|---|
| Ours | **0.892** | **0.60** | **0.46** | **0.60** |
| DPO | 0.591 | 0.17 | 0.28 | 0.18 |
| ReFT | 0.635 | 0.23 | 0.26 | 0.22 |

Table 4: Accuracy of our method and baselines with rejection sampling.

| Datasets | Ours | PPO | DPO | KTO | Skywork |
|---|---|---|---|---|---|
| SMAC | **0.764** | 0.642 | 0.704 | 0.708 | 0.713 |
| MMLU | **0.772** | 0.715 | 0.738 | 0.759 | 0.720 |
| MathQA | **0.804** | 0.732 | 0.765 | 0.773 | 0.731 |

**Rejection Sampling**. We conduct a control study by applying the same rejection sampling procedure from our method to the baselines. For preferred data, we select those where the PROXY LLM can infer the correct decision (called filtered preferred data). This removes possible noisy data. Our method uses this subset, while baselines that require pairwise training data use both this subset and rejected data. As shown in Tab. 4, our method still achieves the best performance under this setting. By contrast, rejection sampling is not guaranteed to improve the performance of other baselines. These results show that the effectiveness of our method is not due to the higher quality of the training data.

**Masked Explicitly Stated Decisions**. We conducted an additional evaluation in which any occurrence of the explicit decision in the generated explanation was masked before passing it to the evaluator model. As shown in Tab 8, this causes only a slight drop in performance. Specifically, our method's accuracy decreases by 2.1% on `SMAC`, 1.8% on `MMLU`, and 3.6% on `MathQA`. We find that in most instances where the explanation states the decision explicitly, the decision can still be inferred from the reasoning trace (e.g., detailed numerical derivations or clear contextual cues). In these cases, masking does not significantly hinder the evaluator's ability to infer the decision. Baselines are also affected under this evaluation protocol, yet our method still outperforms the strongest ones, `KTO` and `ReFT`.

## 5.2 GENERALIZATION TO NEGATIVE SAMPLES: DISTINGUISHING TRUE AND FALSE NEGATIVES

As discussed in Sec. 3.2, we train the rectified flow model $\varphi$ solely on positive samples–explanations that allow the PROXY LLM to infer the ground-truth decision. It is critical that $\varphi$ generalizes to other unseen, negative samples and assigns them reliable rewards. Otherwise, the EXPLANATION

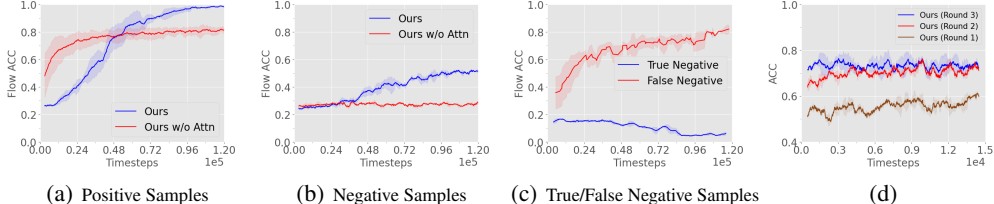

(a) Positive Samples    (b) Negative Samples    (c) True/False Negative Samples    (d)

Figure 2: (a-b) Accuracy of the rectified model $\varphi$ on unseen test samples. Flow ACC is reported as the percentage of samples for which the flow model $\varphi$ assigns the highest logit (reward) to the ground-truth decision. **(c) Our rectified reward model learns to assign low rewards to true negatives—explanations that lead to wrong decisions**—and high rewards to false negatives. Rewards are positively related to accuracy, represented by the y-axis. (d) EXPLANATION LLM accuracy increases with each training round. Each training step corresponds to processing a single training sample.

Table 5: A true negative sample from `MMLU`. Initially, the explanation from the EXPLANATION LLM erroneously supports answer B that focuses on the influence of the filter placement. Post-training with our method, the EXPLANATION LLM accurately justifies the correct answer.

**Context** (shortened): A 67-year-old woman had a pulmonary embolism... She underwent placement of an inferior vena cava (IVC) filter. She had a hematoma that was resolving... Which of the following is the most likely cause of this patient's decreased sensation?

......................................................................................................................................................................

**Decision set** $\mathcal{A}$: [A. Cerebral infarction during the hospitalization; B. Complication of the IVC filter placement; C. Compression of the lateral femoral cutaneous nerve ✓; D. Hematoma of the left thigh.]

......................................................................................................................................................................

**Explanation**: ... The patient's symptoms began after hospitalization, aligning with the placement of the IVC filter... The improvement over time supports a reversible cause related to the filter placement rather than other options... (B, ✗)

......................................................................................................................................................................

**Improved Explanation (Ours):** ...involving nerves near sites of cannula placement or previous bleeding sites... The sensation loss around the left thigh aligns with nerve compression near ✓ the sites of recent interventions.

LLM would not receive informative reward signals. These negative samples fall into two categories. True negatives are invalid explanations rightly rejected by the PROXY LLM, while false negatives are effective explanations misclassified by the PROXY LLM. Effective EXPLANATION LLM training requires distinguishing and differentially rewarding these types. We assess whether our method achieves this separation by first using o1-mini to label negatives as true or false, and then examining the rewards that our algorithm assigns to each type.

Fig. 2(c) demonstrates that our reward model effectively differentiates between the two. Over the course of training, it gradually learns to grant higher rewards to false negatives and lower rewards to true negative samples–precisely the desired behavior. This pattern is mirrored in the accuracy curve: in our framework, rewards are positively related to accuracy, and our reward model achieves approximately 80% accuracy on false negatives and only around 5% on true negatives. To see how this improvement happens, Tab. 5 gives a true negative example where our method successfully guides the EXPLANATION LLM to refine the explanation, an update attributable to correctly computed rewards. Tab. 6 gives an example of a false negative sample. Although the PROXY LLM cannot infer the correct decision, our method is able to deliver correct rewards. We hypothesize that this generalization capability stems from the cross-attention mechanism. Fig. 2 (a) and (b) compare the accuracy of our flow model and the ablation `Ours w/o Attn` on unseen samples. As we can see, without cross-attention, performance on negative samples stagnates throughout training, underscoring its importance.

### 5.3 ABLATION STUDY

**Different Backbone Models**. In Tab. 7, we change the backbone model of our method to Llama-3.1-8B, where our method still surpasses all baselines, achieving a minimum of 2.6% higher ACC on `SMAC`, 2.0% higher ACC on `MMLU`, and 3.7% higher ACC on `MathQA`. We further evaluate the

Table 6: A false negative sample from MMLU where the explanation is correct but the PROXY LLM produces an incorrect distribution. Although the rectified flow model was not trained on this sample, it correctly identifies "the egg roll was present for a substantial time", a cue missed by the PROXY LLM, and thus provides a correct distribution.

---

**Context** (shortened): A wife and her husband were dining at a restaurant owned by a chef. As the wife walked past a table, she slipped on an egg roll that had been on the floor for quite some time, although the chef was unaware it had fallen there. If she sues the chef , she will most likely:

.............................................................................................................................................................

**Decision set $\mathcal{A}$**: [A. Recover, because the egg roll on the floor constituted an unsafe condition of the premises; B. Recover, because the egg roll was on the floor for a substantial period of time before the accident ✓ ; C. Not recover, ...; D. Not recover, ...]

.............................................................................................................................................................

**Explanation**: ... The fact that the egg roll was present for a substantial time ✓ suggests that the owner should have been aware of the potential hazard and taken steps to address it. This situation falls under premises liability where maintaining safe conditions is crucial to ...

.............................................................................................................................................................

**Distribution $p$ from the PROXY LLM:** [**0.9297**, 0.0674, 0.0010, 0.0013] → 'A' ✗
**Distribution $\hat{p}$ from the rectified flow model:** [0.0547, **0.9330**, 0.1089, 0.0685] → 'B' ✓

---

robustness of our method using two more backbone models: Qwen2.5-7B-Instruct, and Gemma-2-2B-It (abbreviated as Qwen and Gemma). As shown in Tab 9, our method consistently improves ACC by up to 19.5% over SFT, confirming its effectiveness across diverse LLM backbones.

**Sparse Reward vs Sentence-level Reward**. As shown in Tab. 10, Ours (Sp Reward) represents the sparse-reward variant and PPO (Se Reward) represents PPO with sentence-level reward. Using sparse reward in our method leads to at least 1.2% lower ACC, indicating that the sentence-level reward design offers better guidance for learning high-quality explanations. With sentence-level reward, PPO still lags behind Ours by at least 5.2% ACC, underscoring the effectiveness of our rectified flow model.

Table 7: Accuracy of our method and baselines with a different backbone model, Llama-3.1-8B.

| Datasets | Ours | SFT | PPO | DPO | KTO | CoT+SFT | ReFT |
|---|---|---|---|---|---|---|---|
| SMAC | **0.689** | 0.583 | 0.632 | 0.650 | 0.655 | 0.614 | 0.663 |
| MMLU | **0.724** | 0.638 | 0.682 | 0.698 | 0.673 | 0.653 | 0.704 |
| MathQA | **0.746** | 0.626 | 0.662 | 0.683 | 0.709 | 0.631 | 0.695 |

Table 8: Accuracy of our method and baselines with masked explicit decisions.

| Datasets | Ours | KTO | ReFT |
|---|---|---|---|
| SMAC | **0.743** | 0.702 | 0.696 |
| MMLU | **0.754** | 0.709 | 0.725 |
| MathQA | **0.768** | 0.732 | 0.735 |

Table 9: Accuracy of our method with another two backbone models.

| Datasets | Llama (Default) | | Qwen | | Gemma | |
|---|---|---|---|---|---|---|
| | SFT | Ours | SFT | Ours | SFT | Ours |
| SMAC | 0.582 | **0.764** | 0.594 | **0.783** | 0.536 | **0.731** |
| MMLU | 0.689 | **0.772** | 0.693 | **0.789** | 0.653 | **0.740** |
| MathQA | 0.667 | **0.804** | 0.645 | **0.778** | 0.626 | **0.723** |

Table 10: Ablations. (Sp Reward): Sparse reward; (Se Reward): Sentence reward.

| | SMAC | MMLU | MathQA |
|---|---|---|---|
| Ours | **0.764** | **0.772** | **0.804** |
| Ours (Sp Reward) | 0.752 | 0.757 | 0.773 |
| PPO (Se Reward) | 0.688 | 0.720 | 0.734 |
| Ours w/o Attn | 0.662 | 0.721 | 0.728 |

**Without Attention**. As shown in Tab. 10, Ours w/o Attn underperforms Ours with a minimum of 5.1% lower ACC, indicating that the transformer-based architecture can better capture the complex context information. Fig. 2(a) and Fig. 2(b) further elucidate the reason: it cannot even reproduce 100% accuracy on positive samples and cannot generalize to negative samples.

Appendix H reports additional results under noisy reward, varying $\sigma$ and excluding SFT.

## 6 CONCLUSION

We demonstrate that a flow-matching generative model can provide sentence-level and reliable rewards for training LLMs to recover a specific policy and explain that policy's decisions in both RL agents and other LLMs. Looking ahead, we envision extending this approach into a general-purpose LLM training framework that automatically generates high-quality reward signals.

## 7 ETHICS STATEMENT

To assess the quality of the synthetic dataset generated by o1-mini, we enlist 12 participants (2 females, 10 males, aged 20-40) to evaluate whether the samples are reasonable and human-like. For the human evaluation, we recruited 45 participants (30 male, 15 female, aged 20–40) from diverse fields, including engineering, chemistry, machine learning, and economics. To ensure consistency, we provided standardized instructions and illustrative examples before the evaluation. All participant information was anonymized, and no personally identifiable data were collected, ensuring that there was no risk to privacy.

## 8 REPRODUCIBILITY STATEMENT

We have undertaken comprehensive efforts to ensure the reproducibility of this work. The complete source code is provided in the supplementary materials. For theoretical analysis, detailed explanations of underlying assumptions and full proofs of the claims are presented in Appx. D. A complete and transparent description of the data processing steps employed in our experiments is also included in the supplementary materials.

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

## A    MORE RELATED WORKS

**Diffusion in Transformer** (DiT, (Yang et al., 2023)) leverages the strengths of self-attention of Transformers to improve the performance of diffusion models across a range of tasks, including image and text generation (Cao et al., 2024). (Dhariwal & Nichol, 2021) demonstrate how Transformer-based architectures can optimize the denoising process in diffusion models, resulting in high-quality image synthesis. (Ulhaq & Akhtar, 2022) explore efficient implementations for diffusion within Transformer. These works are related to our work, as we embed flow matching into the last layer of an LLM. The difference is that we train the LLM-augmented flow matching model as a reward model for the purpose of LLM post-training.

**Cross-attention** is a popular technique for processing information across multiple modalities (Radford et al., 2021; Alayrac et al., 2022; Li et al., 2023). Approaches such as T2I-Adapter (Mou et al., 2024) and VMix (Wu et al., 2024a) use cross-attention mechanisms between text encoders (an LLM) and diffusion models to enhance the generation of high-quality images from textual descriptions. More generally, cross-attention has helped solve tasks that require both vision and language understanding (Hatamizadeh et al., 2025; Cao et al., 2024). Different from previous work on DiT and cross-attention-based image/video generation, to our best knowledge, the proposed method is the first to use generative models and cross-attention to generate rewards for RL-based LLM training.

**Explainable AL/RL without natural language explanations**. Our method is suited within the domain of explainable AI (Arrieta et al., 2020; Carvalho et al., 2019; Ehsan et al., 2019; Gunning, 2017; Ras et al., 2018; Gilpin et al., 2018). *Faithfulness* (Jacovi & Goldberg, 2020) is one of key concepts from the XAI literature, which is defined as the degree to which an explanation reflects the model's actual decision-making process, rather than merely presenting a narrative that sounds plausible to humans (i.e., plausibility).

Recent studies have explored how to generate faithful explanations from large language models. Some approaches (Lyu et al., 2023; Radhakrishnan et al., 2023) focus on enhancing faithfulness through Chain-of-Thought prompting, breaking down the reasoning process into multiple sub-sentences. Others (Schimanski et al., 2024) improve faithfulness by augmenting the training data to provide stronger supervisory signals. For measuring rational faithfulness, Matton et al. (2025) introduces a causal counterfactual evaluation framework using an auxiliary LLM and a Bayesian hierarchical model. Parcalabescu & Frank (2023) proposes CC-SHAP, a token-level self-consistency measure that records model contributions. Madsen et al. (2024) measures self-explanations and find that faithfulness varies significantly across models and datasets. Our method captures the essential behavior of the model by faithfully reproducing its outputs from the given inputs, thereby serving as an interpretable intermediate representation of model behavior. In this sense, our approach offers a form of faithfulness from a functional and behavioral perspective.

For explainable RL (XRL), our approach draws particular parallels with it. *Post-hoc XRL* methods focus on relating inputs and outputs of a trained RL policy in an interpretable way, using an interpretable *surrogate* model as policy approximation. Examples of surrogate models include imitation learning (Abbeel & Ng, 2004), learning from demonstration (Argall et al., 2009), finite state machines (Koul et al., 2018; Danesh et al., 2021), decision trees and their variants (Bastani et al., 2018; Jhunjhunwala, 2019; Bewley & Lawry, 2021; Liu et al., 2018), genetic programming (Zhang et al., 2020), programmatic policy searching (Verma et al., 2018), and deterministic finite automata (Hasanbeig et al., 2021). However, in order to be interpretable, surrogate models are designed as simple as possible. *Ad-hoc XRL* methods represent policies as inherently interpretable models. For example, Silva et al. (2019); Topin & Veloso (2019); Hein et al. (2018); Landajuela et al. (2021) use decision trees as policy approximators. However, the capacity of these models is typically limited. Saliency maps distinguish observation elements that influence decisions (Atrey et al., 2019; Greydanus et al., 2018; Gottesman et al., 2020), but does not capture the reasoning behind decisions (Atrey et al., 2019), leaving humans to give ad-hoc explanations based on these visual cues. Example-based methods use similar experiences to explain decisions (Amir & Amir, 2018; Huang et al., 2018; Zahavy et al., 2016; Topin & Veloso, 2019), but largely overlook underlying reasoning.

**Use explanations to train LLMs**. Various prior works have explored training or tuning language models with explanatory cues, often relying on techniques such as span or word highlighting (Hase & Bansal, 2021; Zhou et al., 2020; Narang et al., 2020; Rajani et al., 2019). Moreover, explicitly training language models with natural language explanations or instructions has been shown to enhance their

ability to leverage explanations in prompts (Wei et al., 2021). These works are different from our method, because we do not use explanations to train LLMs. Instead, we train LLMs to generate explanations.

## B  NOTATIONS FOR TRANSFORMERS

We consider an LLM with $L$ layers, the hidden state after $l$ layers, $\boldsymbol{H}^{(n,l)}$, is projected by three weight matrices $\boldsymbol{W}_Q$, $\boldsymbol{W}_K$, and $\boldsymbol{W}_V$ to the query, key, and value embeddings $\boldsymbol{Q}^{(n,l)}$, $\boldsymbol{K}^{(n,l)}$, and $\boldsymbol{V}^{(n,l)}$, respectively. The self-attention is calculated as:

$$(\boldsymbol{Q}^{(n,l)}, \boldsymbol{K}^{(n,l)}, \boldsymbol{V}^{(n,l)}) = \boldsymbol{H}^{(n,l)}(\boldsymbol{W}_Q, \boldsymbol{W}_K, \boldsymbol{W}_V),$$

$$\boldsymbol{A}^{(n,l)} = \boldsymbol{Q}^{(n,l)} \boldsymbol{K}^{(n,l)^\top} / \sqrt{d_K},$$

$$\text{Attn}(\boldsymbol{H}^{(n,l)}) = \sigma(\boldsymbol{A}^{(n,l)}) \boldsymbol{V}^{(n,l)},$$

where $\sigma(\cdot)$ is SoftMax, and $\boldsymbol{A}$ is the self-attention matrix.

## C  LEARNING FRAMEWORK

**EXPLANATION LLM**. Our method is developed around the EXPLANATION LLM. Given a set $\mathcal{D}_e = \{(a_j, c_j)\}_{j=1}^J$ of decisions $a_j$ and their context $c_j$, we use the following prompt $x_j^e$ to ask the EXPLANATION LLM to generate explanations: Given $\big[$Context c$_{\text{j}}\big]$. Please analyze reasoning for the agent decision based on the context.

Suppose that the EXPLANATION LLM generates $K_e$ sentences as output: $y_j^e = (s_j^{e(1)}, \cdots, s_j^{e(K_e)}])$. For the generated content to be effective explanations, we hope that one can consistently infer agent decisions from explanations across various contexts. To this end, we seek feedback regarding how likely the actual decision $a_j$ is a plausible outcome given each incremental portion of the explanation.

Such feedback is most accurate when provided by human annotators. However, human feedback is expensive (Bai et al., 2022b). We first discuss an alternative approach that uses a PROXY LLM as a surrogate feedback provider, whose downside will motivate the proposed generative reward method introduced in the next subsection.

We query the PROXY LLM with the prompt: Given [Context c$_{\text{j}}$], the reasoning is $s_j^{e(1:k)}$. Thus, the decision is [a decision $a \in \mathcal{A}$]. Here, $s_j^{e(1:k)}$ is the first $k$ sentences of the explanation $y_j^e$. We denote this prompt to the PROXY LLM by $x_j^{g(k)}(y_j^e)$. The dependence on $y_j^e$ will be omitted when unambiguous.

We are interested in the likelihood a decision $a \in \mathcal{A}$ appearing at the end of $x_j^{g(k)}(y_j^e)$, influenced by logits $M(a|x_j^{g(k)}(y_j^e))$ (Eq. 2). In practice, $a$ is represented by some tokens describing the decision. If it involves multiple tokens, we calculate their mean (Yang et al., 2022).

By applying the SoftMax operation to $M(a|x_j^{g(k)}(y_j^e))$, $a \in \mathcal{A}$, we get a distribution over decisions:

$$p(a_j|x_j^{g(k)}) = \text{SOFTMAX}_j \left( M(a|x_j^{g(k)}) \right). \tag{4}$$

This distribution is defined for the first $k$ sentences, $k = 1, \cdots, K_e$. Intuitively, the distribution $p$ changes as we feed the sentences in the explanation $y_j^e$ incrementally. These changes measure the contribution of each sentence to the effectiveness of the explanation, allowing us to define per-sentence rewards for $y_j^e$ as the changes of the likelihood of the actual decision $a_j$ after each newly added sentence:

$$r(s_j^{e(k)}) = p(a_j|x_j^{g(k)}) - p(a_j|x_j^{g(k-1)}), \tag{5}$$

which can be understood as an information gain (Ton et al., 2024). Calculating sentence-level rewards is a trade-off (Team, 2025). We benefit from denser reward signals compared to a single reward for the whole explanation, and also avoid the costs of per-token reward calculation.

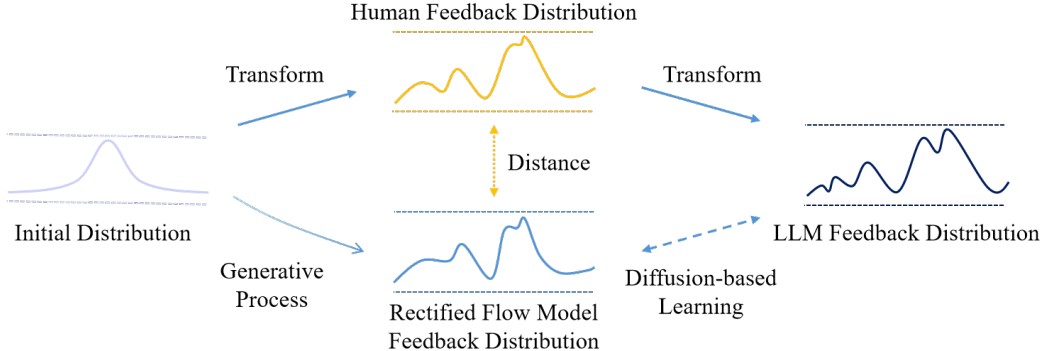

Figure 3: Overview of the idea of our theoretical analysis for the proposed flow-matching–based reward generation framework. We train a flow-matching model to approximate the distribution of LLM-generated feedback. Assuming the LLM feedback distribution is a transformation of the ground-truth human feedback distribution, we derive a closed-form expression for the distance between the induced flow-matching reward distribution and the human feedback distribution.

We intentionally exclude the actual decision $a_j$ from the prompt $x_e$ to the EXPLANATION LLM. Otherwise, the rewards $r(s_j^{e(k)})$ might be trivial and encourage merely restating the decision: the sentence that discloses the decision will get a very large reward, while the following sentences get fairly small rewards, regardless of their content.

The disadvantage of directly using this PROXY LLM is that the rewards $r(s_j^{e(k)})$ could be noisy or inefficient (Yang et al., 2024b), as proven by a baseline method `Proxy LLM` in Tab. 2. We propose to fix this problem by introducing a rectified flow model for reward generation.

**Rectified Flow Model Architecture**.

*Cross-Attention.* $(W_Q, W_K, W_V)$ are last-layer attention weights of the EXPLANATION LLM. We apply these weights to the flow-token hidden states $(H_{EMB})$ to obtain corresponding queries, keys, and values: $(Q_{EMB}, K_{EMB}, V_{EMB}) = H_{EMB}(W_Q, W_K, W_V)$. In parallel, we also obtain the last-layer queries, keys, and values from the EXPLANATION LLM for the given context (i.e., the query and its explanation), denoted as $Q^{(L-1)}, K^{(L-1)}, V^{(L-1)}$, where L is the number of transformer layers. We then perform standard multi-head attention using the concatenated queries, keys, and values $([Q^{(L-1)}, Q_{EMB}], [K^{(L-1)}, K_{EMB}], [V^{(L-1)}, V_{EMB}])$. Finally, we extract the attention hidden states at the flow-token positions as the output.

*Projector.* Define $\boldsymbol{h}_{\text{ATTN}, \boldsymbol{z}_t}$ as the latent state of the flow token $\boldsymbol{h}_{\text{EMB}, \boldsymbol{z}_t}$ after cross-attention. This state has incorporated the ODE time $t$ and explanatory information through cross-attention. Progressing from this state, we use a four-layer fully-connected network $\varphi_{\text{PROJ}} : \mathbb{R}^d \to \mathbb{R}^{|\mathcal{A}|}$ with ReLU activation and layer normalization to generate the vector field $\varphi(t, \boldsymbol{z}_t) = \varphi_{\text{PROJ}}(\boldsymbol{h}_{\text{ATTN}, \boldsymbol{z}_t})$. We find that skip-layer connections are important for training stability. Specifically, we append the inputs $\boldsymbol{z}_t$ and $t$ to the hidden layers of $\varphi_{\text{PROJ}}$.

## D   RECTIFIED FLOW CAN CORRECT ERRORS

**Theorem 1.** *[Error bound for recovering the human reward distribution with a rectified flow] The flow is trained so that the fitting error to the proxy LLM reward distribution, measured in quadratic Wasserstein distance, satisfies $W_2(\hat{p}, q_1) \le \varepsilon$. $p_{\text{FLOW}} := \phi_{1\sharp}p_0$ is the denoised candidate obtained by pushing $p_0$ through the learned flow. Errors relative to the true human distribution are bounded by*

$$W_2(p_{\text{FLOW}}, p) \le \varepsilon + L\sqrt{|\mathcal{A}|}\,\big|\sigma - \sigma_r\big|,$$

*where $L = \exp\big(\int_0^1 \|\nabla\varphi_t\|_\infty\, dt\big)$ is the Lipschitz constant. In many architectures, $L \approx 1$.*

*Proof.* Recall that for any two probability measures $\mu, \nu$ on $\mathbb{R}^d$, the quadratic Wasserstein distance is

$$W_2^2(\mu, \nu) = \inf_{\gamma \in \Gamma(\mu,\nu)} \int_{\mathbb{R}^d \times \mathbb{R}^d} \|x - y\|^2 \, d\gamma(x, y),$$

where $\Gamma(\mu, \nu)$ is the set of all couplings of $\mu$ and $\nu$.

Moreover, if $f : \mathbb{R}^d \to \mathbb{R}^d$ is $L$-Lipschitz then

$$W_2\big(f_\sharp\mu, \; f_\sharp\nu\big) \; \leq \; L \, W_2(\mu, \nu).$$

We want to bound $W_2\big(p_{\text{FLOW}}, \, p\big)$. By the triangle inequality,

$$W_2(p_{\text{FLOW}}, \, p) \; \leq \; W_2\big(p_{\text{FLOW}}, \, q_1\big) \; + \; W_2\big(q_1, \, \hat{p}\big) \; + \; W_2\big(\hat{p}, \, p\big).$$

The middle term is exactly the training error: $W_2(q_1, \hat{p}) \leq \varepsilon$. For the last term, $W_2(\hat{p}, p)$, we observe the fact that $\hat{p} = p * p_0$ shares its mean with $p$; since the corruption is mean-zero Gaussian, the optimal coupling between $p$ and $\hat{p}$ is to pair each clean sample $x$ with $x + \xi$ where $\xi \sim p_0$, giving zero squared cost in expectation.

Hence it suffices to bound the first term, $W_2(p_{\text{FLOW}}, q_1)$.

Since

$$p_{\text{FLOW}} \; = \; \phi_{1\sharp} p_0, \qquad q_1 \; = \; \phi_{1\sharp} q_0,$$

and $\phi_1$ is $L$-Lipschitz by construction,

$$W_2\big(p_{\text{FLOW}}, \, q_1\big) = W_2\big(\phi_{1\sharp} p_0, \, \phi_{1\sharp} q_0\big) \; \leq \; L \, W_2\big(p_0, \, q_0\big).$$

For two zero-mean Gaussians in $\mathbb{R}^{|\mathcal{A}|}$, $\mathcal{N}(0, \sigma_r^2 I)$ and $\mathcal{N}(0, \sigma^2 I)$, one shows

$$W_2\big(p_0, \, q_0\big) = \sqrt{|\mathcal{A}|} \, \big|\sigma_r - \sigma\big|.$$

Combining,

$$W_2(p_{\text{FLOW}}, \, p) \; \leq \; L \, \sqrt{|\mathcal{A}|} \, \big|\sigma - \sigma_r\big| \; + \; \varepsilon,$$

which is exactly the claimed bound. $\qquad\square$

**Lemma 2.** *[Error bound without a rectified flow] Without a rectified flow model, the error of a learning model approximating $\hat{p}$ relative to the true human distribution is bounded by*

$$W_2(\tilde{p}, p) \; \leq \; \varepsilon' \; + \; \sqrt{|\mathcal{A}|} \, \big|\sigma_r\big|,$$

*where $\varepsilon'$ is the model's approximation error with respect to $\hat{p}$. Moreover, the inherent error in the proxy LLM reward distribution is $W_2(\hat{p}, p) = \sqrt{|\mathcal{A}|} \, \big|\sigma_r\big|$.*

*Proof.* First, by the definition of $\varepsilon'$ as the model's approximation error to $\hat{p}$, we have

$$W_2(\tilde{p}, \, \hat{p}) \; \leq \; \varepsilon'.$$

Next, apply the triangle inequality for the 2-Wasserstein distance:

$$W_2(\tilde{p}, \, p) \; \leq \; W_2(\tilde{p}, \, \hat{p}) \; + \; W_2(\hat{p}, \, p).$$

Combining with the above bound gives

$$W_2(\tilde{p}, \, p) \; \leq \; \varepsilon' \; + \; W_2(\hat{p}, \, p).$$

It remains only to compute $W_2(\hat{p}, p)$. Recall that $\hat{p} = p * p_0$ where $p_0 = \mathcal{N}(0, \sigma_r^2 I_{|\mathcal{A}|})$. Since $p$ and $\hat{p}$ differ by the addition of a zero-mean Gaussian of covariance $\sigma_r^2 I_{|\mathcal{A}|}$, one shows directly for two zero-mean Gaussians in $\mathbb{R}^{|\mathcal{A}|}$ that

$$W_2(\hat{p}, p) = \sqrt{|\mathcal{A}|}\,|\sigma_r|.$$

Hence

$$W_2(\tilde{p}, p) \leq \varepsilon' + \sqrt{|\mathcal{A}|}\,|\sigma_r|,$$

as claimed. $\qquad\square$

### D.1 WITH DIFFERENT FUNCTIONAL FORM

The following is the general case where the initial distribution $q_0$ and the error model $p_0$ follow different laws.

For the true distribution $p$ and the flow model's output $p_{\text{flow}}$, the error is bounded by

$$W_2(p_{\text{flow}}, p) \leq \epsilon + \sqrt{\operatorname{tr}\Sigma_{p_0}} + L\sqrt{|\mu_{p_0}|^2 + \operatorname{tr}\Sigma_{p_0} + \operatorname{tr}\Sigma_{q_0}},$$

where $\Sigma_{\cdot}$ denotes the covariance matrix and $\mu_{\cdot}$ is the mean. When both $p_0$ and $q_0$ are Gaussian, this reduces exactly to Theorem 1.

The human-feedback distribution is unknown, but we assume we observe a noisy proxy provided by the LLM with an additive noise model $p_0$. In the proof, the flow network acts as a denoiser: it transforms the proxy toward the latent human distribution. The inequality above quantifies how closely the denoised distribution can approximate the true one.

## E PROMPTS

Below are the prompts used for the EXPLANATION LLM and the PROXY LLM on the SMAC datasets. For the EXPLANATION LLM, the prompt describes the overview of the scenarios along with the previous action-state pairs and instructs the EXPLANATION LLM to analyze the provided information. In response, the EXPLANATION LLM generates a summary of the preceding trajectories and implies the action selection. For the PROXY LLM, the prompt includes a statement summarizing both the input to and output from the EXPLANATION LLM, followed by the phrase: *'Therefore, the missing action is a possible action from $\mathcal{A}$'*. Then, we calculate the mean token logits for each possible action in $\mathcal{A}$, which are then used to determine the action with the highest likelihood.

An example that the prompts used for the EXPLANATION LLM and the PROXY LLM on the
SMAC datasets

**User for EXPLANATION LLM:**
Q: Here are 5 allies and 6 enemies in SMAC. The goal of each ally is to attack all the enemies
together with other allies. At every step, each ally can choose one action to take from the
Action set. The trajectory of each ally consists of a series of State, Action, Value pairs. Here
are 4 consecutive State, Action, Value pairs where the last state-action pair misses the Action:
xxx. Please help me concisely summarize and analyze this part of the trajectory for the
missing Action prediction.

**EXPLANATION LLM:**
R: Throughout the provided trajectory, the ally consistently maneuvered strategically while
maintaining full health and ensuring weapon readiness. Initially, moving SOUTH likely
aimed to xxx. The subsequent WEST movement improved xxx. **The predicted next action
should be a directional movement aimed at advancing toward a more optimal attacking
position. Compared to other directions, EAST offers the best alignment for coordinated
attacks, ensuring allies can effectively converge on and target enemies together.**

**User for PROXY LLM:**
Q + R + Therefore, the missing action is {a possible action from $\mathcal{A}$}.

**PROXY LLM:**
The token logits of the given possible action.

An example that the prompts used for the EXPLANATION LLM and the PROXY LLM on the
AI2-THOR datasets

**User for EXPLANATION LLM:**
Q: There are multiple robots in the simulation,each with different skills and capabilities. The
robots are tasked with performing a series of actions in a simulated environment to complete
a certain task. Robot one xxx. Task is xxx. Current environment condition is xxx. The
previous actions include xxx. The available actions are xxx. You act as an AI assistant to
help the robots decompose the tasks into actions step by step.

**EXPLANATION LLM:**
R: Toggling on the television and placing the keychain on the coffee table. Robot 0 now
should take a corrective navigation step **(GoToObject)** before successfully executing
ToggleOnObject. After completing this sub-task, Robot 1 should navigate to the coffee table
and keychain, pick up the keychain, and place it on the table in the correct order.

**User for PROXY LLM:**
Q + R + Therefore, the next action is {a possible action from $\mathcal{A}$}.

**PROXY LLM:**
The token logits of the given possible action.

Above are the prompts used for the EXPLANATION LLM and the PROXY LLM on the AI2-THOR
datasets. For the EXPLANATION LLM, the prompt describes the overview of the scenarios along
with the previous action pairs and instructs the EXPLANATION LLM to help robots to take action
step by step. In response, the EXPLANATION LLM generates potential trajectories and implies the
action selection. For the PROXY LLM, the prompt includes a statement summarizing both the input
to and output from the EXPLANATION LLM, followed by the phrase: *'Therefore, the next action is*

*a possible action from $\mathcal{A}$'*. Then, we calculate the mean token logits for each possible action in $\mathcal{A}$, which are then used to determine the action with the highest likelihood.

---

An example that the prompts used for the EXPLANATION LLM and the PROXY LLM on the MMLU datasets

**User for EXPLANATION LLM:**
Q: Below is a multiple-choice question about professional law. Please explain your reasoning without explicitly revealing your selection. The question is: Which of the following is not within the power of a federal subpoena to command a person to do at a specified time and place? A: Submit to a physical or mental examination.; B: Attend and testify as to persons and/or events related to a cause of action.; C: Produce designated documents, electronically stored information, or tangible things in that person's possession, custody, or control.; D: Permit the inspection of premises.

**EXPLANATION LLM:**
R: Federal subpoenas are designed to compel testimony, the production of documents, and permit inspections related to a case. However, requiring an individual to undergo a physical or mental examination typically falls outside the scope of what a subpoena can mandate. **Such examinations usually require separate legal processes or warrants, ensuring individuals' rights are adequately protected.**

**User for PROXY LLM:**
Q + R + Therefore, the correct answer is {a possible answer from $\mathcal{A}$}.

**PROXY LLM:**
The token logits of the given possible answer.

---

An example that the prompts used for the EXPLANATION LLM and the PROXY LLM on the MathQA datasets

**User for EXPLANATION LLM:**
Q: Below is a multiple-choice question about math problem. Please explain your reasoning without explicitly revealing your selection. The question is: an auction house charges a commission of 18 % on the first $50,000 of the sale price of an item, plus 10 % on the amount of the sale price in excess of $50,000. what was the price of a painting for which the house charged a total commission of $24,000? a ) $115,000, b ) $160,000, c ) $ 200,000, d ) $ 240,000, e ) $365,000

**EXPLANATION LLM:**
R: **Say the price of the house was $ x, then 0.18 * 50,000 + 0.1 * ( x - 50,000 ) = 24,000 $\rightarrow$ x = $ 200,000** ( 18 % of $ 50,000 plus 10 % of the amount in excess of $ 50,000, which is x - 50,000, should equal to total commission of $ 24,000 )

**User for PROXY LLM:**
Q + R + Therefore, the correct answer is {a possible answer from $\mathcal{A}$}.

**PROXY LLM:**
The token logits of the given possible answer.

---

There are the prompts used for the EXPLANATION LLM and the PROXY LLM on the MMLU and MathQA datasets. For the EXPLANATION LLM, the user provides the question along with the possible answer options and requests the EXPLANATION LLM to explain the reasoning process without explicitly revealing the selection. In response, the EXPLANATION LLMgenerates an explanation that

analyzes the problem and implicitly suggests an answer. For the PROXY LLM, the user presents a statement that includes both the input to and the output from the EXPLANATION LLM, followed by the phrase: *'Therefore, the correct answer is a possible answer from $\mathcal{A}$'.* to calculate the mean token logits of the given possible answer. Then, we calculate the mean token logits for each possible answer in $\mathcal{A}$, which are then used to determine the answer with the highest likelihood.

## F  ADDITIONAL EXPERIMENTAL SETUP

**Benchmark**.

**(1)** SMAC. We generate explanations for an ally agent based on action-state history. Our dataset consists of 2K trajectories (1.5K for training and 0.5K for evaluation), with each trajectory containing states, actions, and rewards for 30 time steps. We feed information of previous 4 consecutive time steps to the EXPLANATION LLM to generate explanations.

Specifically, we first train an RL policy using MAPPO (Yu et al., 2022) on SMAC and then collect the action-state pairs at each timestep during the evaluation phase. Each action-state pair contains information about the current agent's position, health, and weapon cooldown, as well as the corresponding details for visible allies and enemies. Additionally, the action selected by the current agent is included. All action-state pairs are stored in JSON format:

$$State : \{visible\ allies : \{relative\ (x, y) ; weapon\_cooldown; health\}$$
$$\{id0 : [0.0765, -0.0765] ; 0.0; 1.0\} ; \{id1 : [0.0765, 0.0] ; 0.0; 1.0\} ;$$
$$\{id2 : [0.0765, 0.0765] ; 0.0; 1.0\} ; \{id3 : [0.153, 0.0] ; 0.0; 1.0\} ;$$
$$None\ visible\ enemies; own\_health : 1.0\ \} ; Action : SOUTH$$

We feed the EXPLANATION LLM with 4 consecutive action-state pairs, masking the action in the final pair, and then instruct the EXPLANATION LLMto analyze the provided information and infer the missing action.

**(2)** AI2-THOR. We evaluate our method on AI2-THOR, an embodied AI benchmark widely used in RL research, thereby testing the proposed method in a fully sequential decision-making environment.

Specifically, AI2-THOR includes household activity scenarios that require multi-step planning, multi-robot coordination, and interaction with dynamic environments. We use the *Complex Tasks* split from the SMART-LLM dataset, in which each task involves 2–4 heterogeneous robots with complementary skills. These tasks are challenging because they cannot be decomposed into independent sub-problems solvable by a single robot; agents must reason over past actions and observations, perform long-horizon planning, and coordinate multiple robots strategically to leverage their combined capabilities. This setting demands strong reasoning and sequential decision-making, making it a suitable benchmark for evaluating our method's ability to recover a complex underlying policy. The original Complex Tasks split contains only 8 tasks. To broaden task diversity, we expand the set to 59 tasks by prompting GPT-4o to generate additional tasks that follow the same structural requirements.

The prompt is formatted as:

*AI2THOR is a embodied-AI benchmark xxx.*
*The robots should be xxx.*
*The tasks should be xxx.*
*Here are some examples of embodied-AI task.*
*Example 1: xxx.*
*...*
*Example n: xxx.*
*Please help me generate new tasks that satisfy the above requirement.*
After generating these new tasks, we ask both human and GPT-4o to verify the correctness of these tasks.

We then apply RL to obtain a near-optimal policy over these 59 tasks and record its successful trajectories of 70% tasks for training our method and baselines. During evaluation, we measure the average success rate of our method and baselines on the remaining 30% unseen tasks to assess their generalization capability.

**(3)** `MMLU`. We choose 4 challenging Professional Knowledge subsets (Professional214 Medicine (272 samples), Professional Law (1.53K samples), Professional Accounting (282 samples),215 and Professional Psychology (612 samples)). For each dataset, 70% of the samples are selected randomly for training, and the remaining 30% for evaluation.

**Training Details**. For training dataset construction, established instruction-tuning datasets provide (general-purpose) preference data for answers, but we optimize explanations for answers. As a result, existing instruction-tuning datasets can't perform very well in our tasks. Constructing of a human-annotated dataset is labor-intensive, so we leverage OpenAI o1-mini to generate explanations for each sample, providing a more efficient alternative. To generate positive samples, o1-mini is prompted with a question and its correct answer. The prompt used for positive sample generation is as follows: *'Below is a multiple-choice question about xxx. Please explain your reasoning without explicitly revealing the correct answer: xxx. The question is: xxx . The options are xxx.'*. Similarly, to generate negative samples, o1-mini receives a question with a randomly selected incorrect answer in the prompt: *'Below is a multiple-choice question about xxx. Please explain your reasoning without explicitly revealing a wrong answer: xxx. The question is: xxx . The options are xxx.'*.

To further assess the quality of the generated samples, we enlist 12 participants (2 females, 10 males, aged 20-40) to evaluate whether the samples are reasonable and human-like. We randomly select 5% randomly chosen samples from each datasets and ask panticipants to rate them on a scale of 1 to 4: 1 = unreadable, 2 = readable but unreasonable, 3 = reasonable but unnatural, and 4 = natural and human-like. The example of a sample is listed below.

---

**An example that the evaluation of the quality of our constructed dataset on `MathQA`**

**Q** = Below is a multiple-choice question about professional law. Please explain your reasoning without explicitly revealing your selection. The question is: At a defendant's trial for drug dealing, a prosecution witness testified that he had heard the defendant telling a group of people to come to the defendant's house because he had "a great crop of apples for sale." The prosecutor then called and qualified a drug enforcement agent as an expert in how drug dealing is conducted. The prosecutor now seeks to have the expert testify that in her opinion, based on years of experience with drug dealers, the defendant's statement about the apples was code for his having drugs for sale. Should the expert's opinion be admitted?

**Decision set $\mathcal{A}$ =**
[**a**: No, because an expert must base his or her opinion on scientific or technical data, not merely on experience.;
**b**: No, because the opinion is based on the criminal actions of others engaged in drug dealing, and thus is substantially more unfairly prejudicial than probative.;
**c**: Yes, because the opinion is based on her specialized knowledge, and it will assist the trier of fact in understanding the evidence and determining facts in issue in the case.;
**d**: Yes, provided that the expert first testifies about the specific information upon which she based her view that the defendant was using "apples" as a code word to refer to drugs.]

**Explanation:** The expert's testimony is likely admissible because her specialized knowledge can clarify the defendant's statements in the context of drug dealing. Courts generally allow experts to provide opinions that assist the jury in understanding evidence or determining facts. As long as her opinion is based on her experience and doesn't rely solely on hearsay, it should help the trier of fact make informed decisions.

Score: (1-4, 1=unreadable, 2=readable but unreasonable, 3=reasonable but unnatural, 4= natural and human-like):

---

We train the EXPLANATION LLM and the rectified flow model iteratively. In the first step, EXPLANATION LLM generates an explanation for each sample 3 times in the training set. The PROXY LLM then classifies these explanations as positive or negative. We use the positive samples to train the rectified flow model. Once trained, the rectified flow model is employed in subsequent the EXPLANATION LLM training. This entire process constitutes one round. As shown in Fig. 2(d), the EXPLANATION LLM's accuracy plateaus during the third round because the updates to the reward

model no longer yield significant performance improvements. By comparison, in the first two rounds, the accuracy initially increases rapidly due to substantial updates to the reward model, followed by a more gradual improvement. Therefore, we perform two rounds of training. Note that the training process can be adapted for online learning, where the EXPLANATION LLM and the rectified flow model are trained simultaneously.

In each round, the EXPLANATION LLMand the flow model are trained for 10 and 100 epochs, respectively. The learning rate for the EXPLANATION LLMis initialized as $2e{-}5$ and then decays linearly to zero, whereas the learning rate for the flow model is fixed at $2e{-}4$. EXPLANATION LLM uses the Adam optimizer, BF16 and LoRA with hyperparameters $r_{lora} = 16$ and $\alpha_{lora} = 16$, while DeepSpeed ZeRO-3 is employed to accelerate training. The rectified flow model uses the Adam optimizer. The batch size is 12 for the EXPLANATION LLM training and 128 for the flow model training. We run our method on 4 parallel 80GB Nvidia A100 GPUs. For EXPLANATION LLM training, each GPU processes at most 1 sample at a time, with gradient accumulation over 3 steps. For the flow model training, each GPU processes at most 32 samples at a time. SFT typically takes 0.5 to 1 hour, while our method requires approximately 14–30 hours on different datasets.

**Evaluation**. For evaluation, we use temperature $\tau = 0.7$ across all settings. The maximum number of generated tokens is 350 for SMAC, and 200 for MMLU and MathQA.

For human evaluation, we recruit 45 participants(30 males, 15 females, aged 20-40) from diverse fields (engineering, chemistry, machine learning, and economics). Each participant completes a questionnaire containing 3 randomly selected benchmark samples. For test samples outside an annotator's expertise, we suggest that annotators consult authoritative sources for unfamiliar terminology or background context to better judge whether the explanations are factually grounded. To ensure consistency across participants, we provided clear instructions and illustrative examples prior to the evaluation. For each task, participants were presented with multiple explanations generated by different methods and asked to make a decision based on each one independently. Immediately afterward, participants selected the explanation they perceived as least cognitively demanding, most logically sound, and most actionable, with the order of explanations randomized for each participant to ensure fairness and minimize bias. This procedure enabled us to analyze trends with statistical confidence.

We define cognitive load as the perceived mental effort required to complete a task. Importantly, actionability and ACC capture distinct concepts. Actionability reflects how easily a third party can select an answer based on an explanation, emphasizing its clarity and usability. By contrast, ACC denotes the accuracy with which a third party can predict the underlying decision based on the explanation. Notably, an explanation may be highly clear (high actionability) yet still lead to an incorrect prediction (low ACC). The score for each metric is computed as the proportion of participants selecting a given method as the best. The example of a sample is listed below. We make sure that all the information about the participants is invisible, and there is no potential risk to their privacy.

An example that the evaluation of the effectiveness of our method on `MathQA`. The explanations generated by `DPO`; `ReFT` and our method are named as A1; A2; A3 and A4 in a random order.

**Description** = Optimal MathQA Problem-Solving Strategies: Choose the Best Explanation: Below is a question along with three explanations (A1, A2, A3). Please review the question and the three explanations carefully, then answer the following questions accordingly

**Q** = A shopkeeper sold an article offering a discount of 5% and earned a profit of 31.1%. what would have been the percentage of profit earned if no discount had been offered? A: 38, B: 27.675, C: 30, D: Data Inadequate, E: None of These.

**A1**: Let's assume that the cost price of the article is 'C'. When the shopkeeper offered a discount of 5%, the selling price of the article became 95% of the original selling price. Let's call the original selling price 'S'. Therefore, the new selling price is 0.95S. We know that the shopkeeper earned a profit of 31.1%. Therefore, we can write the equation: Profit percentage = (Profit / Cost price) x 100. 31.1 = [(0.95S - C) / C] x 100. Now, let's assume that the shopkeeper did not offer any discount. In that case, the selling price would have been the original selling price 'S'. Let's call the new profit percentage 'P'. Therefore, we can write the equation: P = [(S - C) / C] x 100. We need to find the value of 'P' in terms of 'C' and 'S'. From the first equation, we can solve for 'C': C = (0.95S - 0.311S) / 0.311. C = 2.978S. Substituting this value of 'C' in the second equation, we get: P = [(S - 2.978S) / 2.978S] x 100 P = 33.1%.

After reading Explanation A1, which answer do you think it is leading to? A: 38; B: 27.675; C: 30; D: Data Inadequate; E: None of These. F: Not Sure

**A2**: xxx

After reading Explanation A2, which answer do you think it is leading to? A: 38; B: 27.675; C: 30; D: Data Inadequate; E: None of These. F: Not Sure

**A3**: xxx

After reading Explanation A3, which answer do you think it is leading to? A: 38; B: 27.675; C: 30; D: Data Inadequate; E: None of These. F: Not Sure

**A4**: xxx

After reading Explanation A4, which answer do you think it is leading to? A: 38; B: 27.675; C: 30; D: Data Inadequate; E: None of These. F:Not Sure

Which explanation do you think is the most logically sound? A: A1; B: A2; C: A3; D: A4

Which explanation induce the least cognitive load? A: A1; B: A2; C: A3; D: A4

Which explanation is the most actionable? (Knowing which answer to select after reading it.) A: A1; B: A2; C: A3; D: A4

## G  BASELINES

`SFT` is a widely used approach to fine-tune LLMs for specific downstream tasks by training them on labeled datasets. In this process, a pre-trained LLM is exposed to task-specific data containing input-output pairs. The goal is to minimize the difference between the model's predictions and the ground truth, typically using a supervised learning objective like cross-entropy loss. We use a training dataset consisting of preferred examples generated by o1-mini. These examples are effective explanations. We use `SFT` as the initialization stage to enable the EXPLANATION LLM to adhere

to the target format and to set a foundation for subsequent refinement toward the target explanatory behavior. We use BF16 and DeepSpeed ZeRO-2 to accelerate training. The batchsize is 12 and the optimizer is Adam. The training epochs of SFT is 3 and the learning rate is initialized as $2e{-}5$ recommended by the TRL library and then decays linearly to zero.

PPO is a reinforcement learning algorithm commonly used to fine-tune LLMs, enhancing their alignment with specific goals or human preferences. PPO fine-tunes a pre-trained model by optimizing its policy based on feedback signals. These signals typically come from a reward model, which is trained on labeled data that reflects desirable outputs. PPO adjusts the model's parameters to maximize expected reward while constraining updates within a predefined range, avoiding large deviations that could destabilize training. We initialize the model with the same SFT procedure, leveraging the same o1-mini–distilled preferred examples. To generate more effective explanations, we further fine-tune this SFT model with a reinforcement learning algorithm, PPO, which is supposed to reinforce explanations that better support inferring the policy's decision by assigning higher rewards. To achieve this, we train a reward model on a paired preference dataset (queries with corresponding positive/negative explanations). We then fine-tune the SFT model with PPO using this reward model. We use BF16 and LoRA with hyperparameters $r_{lora} = 16$ and $\alpha_{lora} = 16$, while DeepSpeed ZeRO-3 is employed to accelerate training. The batchsize is 12 and the optimizer is Adam. The training epochs of the reward model and PPO are both 10 and the learning rate is initialized as $3e{-}6$ recommended by the TRL library and then decays linearly to zero. The main difference between PPO and our method lies in the reward modeling. Our method trains a probabilistic reward model using flow matching, while PPO uses a standard deterministic reward model trained with the Bradley-Terry model.

DPO is a technique for aligning LLMs with human preferences by directly optimizing their outputs using labeled preference data. Unlike traditional reinforcement learning from human feedback, which relies on a reward model to evaluate responses, DPO simplifies the alignment process by directly using preference comparisons to guide optimization. In DPO, the labeled data consists of paired responses where one option is preferred over the other. The model learns to produce outputs that align with these preferences by optimizing a contrastive objective. For DPO, we initialize the model with the same SFT procedure, leveraging the same preferred examples distilled from o1-mini. To generate more effective explanations, we then fine-tune this SFT model using preference-based learning via DPO. Unlike PPO, DPO does not require a separate reward model. Instead, we use paired preference data (each query paired with a preferred and a dispreferred explanation), freeze the SFT checkpoint as the reference model, and fine-tune the EXPLANATION LLM using the DPO objective. After training converges, we generate explanations using the same prompts as the other baselines. We use BF16 and LoRA with hyperparameters $r_{lora} = 16$ and $\alpha_{lora} = 16$, while DeepSpeed ZeRO-2 is employed to accelerate training. The batchsize is 12 and the optimizer is Adam. The training epochs of DPO is 10 and the learning rate is initialized as $5e{-}6$ recommended by the TRL library and then decays linearly to zero.

KTO is an advanced approach for aligning LLMs with human preferences or specific task objectives. It draws inspiration from prospect theory, a behavioral economics framework that models how humans evaluate potential gains and losses under uncertainty. In this context, KTO optimizes the alignment process by weighting outputs based on their perceived utility, rather than treating all errors equally. The core idea of KTO is to model alignment as an optimization problem where the goal is to maximize expected utility under a prospect-theoretic framework. For KTO, we initialize the model using the same SFT procedure, leveraging the same preferred examples distilled from o1-mini. To further improve explanation quality, we fine-tune the SFT model using preference-based learning via KTO. Unlike PPO, KTO does not require a separate reward model. Instead, we construct a dataset of (query, explanation, label) triples, where the label indicates whether the explanation is preferred or dispreferred. We then optimize the EXPLANATION LLM using the KTO loss, with the SFT checkpoint kept as a frozen reference for regularization. After training, the KTO-tuned model is prompted with the same context as the other baselines to generate explanations. We use BF16 and LoRA with hyperparameters $r_{lora} = 16$ and $\alpha_{lora} = 16$, while DeepSpeed ZeRO-2 is employed to accelerate training. The batchsize is 12 and the optimizer is Adam. The training epochs of KTO is 10 and the learning rate is initialized as $5e{-}7$ recommended by the TRL library and then decays linearly to zero.

Skywork. Skywork-Reward-Llama-3.1-8B-v0.2 is a reward model developed by Skywork AI as part of their efforts to improve alignment and instruction-following abilities in large language models. Based on the LLaMA-3.1 8B architecture, this version—v0.2—is specifically trained to evaluate and

rank responses generated by language models in a preference-based learning setting. The model is fine-tuned using a dataset of paired responses labeled by humans or heuristic methods, where each pair contains a preferred and a less preferred answer. This enables the model to predict a scalar reward score for any given response, facilitating its use in fine-tuning chat models for improved helpfulness, relevance, and alignment with human expectations. It ranks among the top two 8B models on the RewardBench Leaderboard, and all top five 8B models adopt it as their backbone and utilize its preference dataset for training. For `Skywork`, we adopt Skywork-Reward-Llama-3.1-8B-v0.2 as the base model. This checkpoint has already been optimized for reward modeling. We further fine-tune it using `SFT` on the same preferred explanation dataset to adapt it to the explanation generation task. Then, we apply `PPO` using this trained reward model to further align the model's outputs with human preferences. We using BF16 and LoRA with hyperparameters $r_{lora} = 16$ and $\alpha_{lora} = 16$. DeepSpeed ZeRO-3 is employed to accelerate training. The batchsize is 12 and the optimizer is Adam. The training epoch of `SFT` is 3. The training epoch of the reward model and `PPO` is both 10, and the learning rate is initialized as $3e{-}6$ recommended by the TRL library and then decays linearly to zero.

`Deepseek`. DeepSeek-R1-Distill-Llama-8B is a high-performance, open-source language model developed by DeepSeek. This model is a distilled version of DeepSeek-R1, built upon the Llama-3.1-8B-Instruct architecture. By distilling knowledge from the larger DeepSeek-R1 model, it provides state-of-the-art performance with reduced computational requirements, making it more accessible for various applications. For `DeepSeek`, we use DeepSeek-R1-Distill-Llama-8B as the base model and apply `SFT` using the same preferred explanation dataset to adapt it to the explanation generation task. We use BF16 and LoRA with hyperparameters $r_{lora} = 16$ and $\alpha_{lora} = 16$. DeepSpeed ZeRO-3 is employed to accelerate training. The batchsize is 12 and the optimizer is Adam. The training epoch of the `SFT` model is 3 and the learning rate is initialized as $2e{-}5$ recommended by the TRL library and then decays linearly to zero.

`CoT+SFT` is a fine-tuning method that enhances the reasoning capabilities of LLMs by combining `SFT` with the structured reasoning paradigm. CoT uses explicit programmatic representations, such as pseudo-code or structured logic, to model complex problem-solving tasks. In this approach, `SFT` is performed using datasets annotated with both input-output pairs and detailed programmatic reasoning traces. These traces serve as templates for step-by-step reasoning and enable the model to break down complex problems, such as mathematical reasoning or logical inference, into manageable sub-tasks. The explicit program-like structure helps the model perform multi-step computations and enhances interpretability, making it especially useful for domains requiring precision and transparency. For `CoT+SFT`, we construct a `CoT`-augmented version of our preferred explanation dataset via o1-mini by replacing each explanation with a `CoT` variant, containing more explicit step-by-step reasoning. We then apply `SFT` using this modified dataset to the same base model as our method. We adopt the open source library from (Trung et al., 2024), where we use BF16, DeepSpeed ZeRO-2, a fixed learning rate of 1e-5, Adam optimizer and a batchsize of 12.

`ReFT` is a training approach designed to enhance the reasoning capabilities of LLMs by combining supervised fine-tuning (`SFT`) with reinforcement learning. In `ReFT`, the initial training begins with `SFT`, where the model is fine-tuned using datasets annotated with reasoning traces, such as step-by-step explanations or logical chains of thought. Once the model achieves a baseline performance, reinforcement learning is applied to further refine its reasoning capabilities. For `ReFT`, we use the same `COT`-augmented dataset as in `CoT+SFT` and the `ReFT` algorithm to the same base model as our method. We adopt the open source library from (Trung et al., 2024), where we use BF16, DeepSpeed ZeRO-2, a fixed learning rate of 3e-7, Adam optimizer and a batchsize of 12.

`o3-mini` is a smaller, more cost-efficient reasoning model in OpenAI's "o3" family. It's optimized for tasks in STEM domains — especially math, science, and coding — with the goal of higher accuracy, lower latency, and lower cost compared to larger models. We directly use it for evaluation without fine-tuning.

`o3-mini` is used exclusively for inference without fine-tuning, whereas `CoT+SFT` and `ReFT` rely on the `CoT`-enhanced dataset. All other baselines, as well as our method, are trained on the same dataset.

## H  ADDITIONAL RESULTS

### H.1  DIFFERENT CHOICES OF $\sigma$ IN $q_0$

We evaluate the impact of varying $\sigma$ in $q_0$ on final performance across different datasets. As shown in Table 11, the results remain stable across a wide range of values ($\sigma^2 = 0.5, 1, 5$). Specifically, on SMAC, the performance slightly decreases from 0.772 at $\sigma^2 = 0.5$ to 0.761 at $\sigma^2 = 5$, while MMLU exhibits a small improvement from 0.759 to 0.781 as $\sigma^2$ increases. For MathQA, performance is highest at $\sigma^2 = 0.5$ (0.809) and decreases mildly as $\sigma^2$ grows. These variations are minimal, underscoring that our method is not sensitive to the choice of $\sigma$ and thereby confirming the robustness predicted by our theoretical analysis.

Table 11: Performance of our method with different $\sigma$ in $q_0$.

| Datasets | $\sigma^2 = 0.5$ | $\sigma^2 = 1$ | $\sigma^2 = 5$ |
|---|---|---|---|
| SMAC | 0.772 | 0.764 | 0.761 |
| MMLU | 0.759 | 0.772 | 0.781 |
| MathQA | 0.809 | 0.804 | 0.783 |

### H.2  WITHOUT SFT

To evaluate the influence of supervised fine-tuning on our method, we train our method and baselines without conducting SFT. Results are shown in Tab. 12. Our method does require SFT to achieve strong performance, possibly because SFT data is crucial for obtaining enough positive samples to train the rectified flow model. Removing SFT, the performance of our method and baselines drops particularly sharply on RL tasks like SMAC. RL tasks typically involve sequential decision-making and even interactions among multiple agents, making the reasoning of correct actions particularly challenging. SFT offers valuable examples about how to reason in such complex scenarios. Notably, our method without SFT significantly outperforms baselines without SFT on SMAC, demonstrating its effectiveness in cases characterized by higher reasoning demands.

Table 12: Performance of our method and baselines without supervised fine-tuning.

| Datasets | Ours | PPO | DPO | KTO | ReFT | Skywork |
|---|---|---|---|---|---|---|
| SMAC | 0.457 | 0.302 | 0.324 | 0.381 | 0.405 | 0.357 |
| MMLU | 0.605 | 0.586 | 0.590 | 0.621 | 0.602 | 0.608 |
| MathQA | 0.634 | 0.618 | 0.630 | 0.638 | 0.643 | 0.629 |

### H.3  UNDER NOISY REWARD

To evaluate the robustness of our method under noisy reward signals, we conduct an additional experiment where the reward (0/1) of the final token is randomly flipped with probability 0.03. As shown in Tab. 13, under the noisy reward condition, the performance of our method decreases only slightly from 77.2% to 74.3% on unseen tasks in the MMLU benchmark. Notably, even with these noisy rewards, our method still outperforms PPO (72.3%), Skywork (73.7%), and DeepSeek (72.1%) trained without noise. This demonstrates the robustness of our approach under imperfect reward feedback.

Table 13: Performance of our method under noisy reward.

| | SMAC | MMLU | MathQA |
|---|---|---|---|
| Ours (Noise) | \ | 0.743 | \ |

## H.4 Visualized Examples

SMAC. To showcase the effectiveness of our method, we visualize an example on SMAC that a negative explanation classified by PROXY LLM is corrected by the rectified flow model $\varphi$. As shown above, this explanation infers the answer is related to 'Attack Enemy 0', however, the PROXY LLM fails to predict the correct action. Instead, the rectified flow model $\varphi$ can predict the action correctly based on the explanation.

Besides, we visualize another example that a negative explanation generated by EXPLANATION LLM is optimized to a positive explanation after training for two rounds. As shown below, the explanation from the SFT model opts for the movement of EAST. The explanation of our method can infer the correct movement of NORTH with solid evidence.

MathQA. To showcase the effectiveness of our method, we visualize an example on MathQA that a negative explanation classified by PROXY LLM is corrected by the rectified flow model $\varphi$. As shown below, this explanation infers the answer is related to option d, however, the PROXY LLM fails to predict the correct answer. Instead, the rectified flow model $\varphi$ can predict the answer correctly based on the explanation.

Besides, we visualize another example on MathQA that a negative explanation generated by EXPLANATION LLM is optimized to a positive explanation after training for two rounds. As shown below, the explanation from the SFT model is confusing and misleading. The explanation of our method can infer the correct answer c with solid evidence.

---

An example that a negative explanation classified by the PROXY LLM is corrected by the rectified flow model $\varphi$ of on SMAC

**Q** = Here are 5 allies and 6 enemies in SMAC. The goal of each ally is to attack all the enemies together with other allies. At every step, each ally can choose one action to take from the Action set. The trajectory of each ally consists of a series of State, Action, Value pairs. Here are 4 consecutive State, Action, Value pairs where the last state-action pair misses the Action: xxx. Please help me concisely summarize and analyze this part of the trajectory for the missing Action prediction.

**Action set** $\mathcal{A}$ = ['DEAD', 'STOP', 'NORTH', 'SOUTH', 'EAST', 'WEST', 'Attack Enemy 0' ✓, 'Attack Enemy 1', 'Attack Enemy 2', 'Attack Enemy 3', 'Attack Enemy 4', 'Attack Enemy 5']

**Explanation:** ...In the first state, Ally opts to move EAST, likely positioning itself xxx. Ally's next action, "Attack Enemy 3", shows a shift to offensive behavior, ... Ally's following action, "Attack Enemy 0", further prioritizes offense, which might reflect a high-risk engagement. Ally's last action solidifies this pattern, further engaging Enemy 0, likely due to the imminent engagement's hazards. Throughout these actions, ... The allies prioritize attacking specific foes based on their availability and proximity.✓

**Distribution** $p$ **from the PROXY LLM:** [0.0074, 0.0031, 0.0260, 0.0062, 0.0055, 0.0096, 0.1069, 0.2148, **0.2812**, 0.1089, 0.1260, 0.1045] → 'Attack Enemy 2' ✗

**Distribution** $\hat{p}$ **from the rectified flow model:** [0.0084, 0.0040, 0.0084, 0.0061, 0.0064, 0.0099, **0.3254**, 0.1110, 0.1137, 0.1771, 0.1065, 0.1218] → 'Attack Enemy 0' ✓

An example on SMAC that a negative explanation generated by the PROXY LLM is optimized to a positive explanation after training for two rounds

**Q** = Here are 5 allies and 6 enemies in SMAC. The goal of each ally is to attack all the enemies together with other allies. At every step, each ally can choose one action to take from the Action set. The trajectory of each ally consists of a series of State, Action, Value pairs. Here are 4 consecutive State, Action, Value pairs where the last state-action pair misses the Action: xxx. Please help me concisely summarize and analyze this part of the trajectory for the missing Action prediction.

**Actual Action** = NORTH

**Explanation:** In the first state, ally 0,1,2, and 3 all move south, ... The following action, STOP, ... Subsequently, ally 0 and 2 move WEST, ... Throughout these steps, allies prioritize movement over direct engagement, likely because no enemies are in view. Given the current state, where allies are arranged with their backs touching and no Enemy is in sight, the next logical action is EAST. This directional movement likely aims to realign the formation back to its original or a more advantageous state, ensuring that allies are positioned optimally for future engagements. This EAST movement contrasts with WEST, which may have been a short-term repositioning to balance the formation (EAST, ✗), and NORTH and SOUTH, which were likely attempts to regroup or reposition relative to other allies.

**Improved Explanation (Ours):** In the first state, the ally chooses to move SOUTH, ... In the second state, the ally STOPs, ... Moving WEST in the third state likely aims to cover more ground or flank the enemies, ... In the latest state, with no visible enemies and all allies maintaining full health and no weapon cooldowns, the ally opts to move NORTH. This directional movement decision likely stems from a strategic need to reposition for a better vantage point or to avoid potential threats, given the NORTHward movement may offer a more advantageous stance compared to other directions.✓ Choosing NORTH over EAST, for instance, may be due to factors such as better cover or better line-of-sight, enhancing the ally's readiness to engage when enemies become visible.

An example that a negative explanation classified by the PROXY LLM is corrected by the rectified flow model $\varphi$ of on MathQA

**Q** = the cash realised on selling a 14% stock is rs. 101.25, brokerage being 1/4% is.

**Decision set** $\mathcal{A}$ = [a: 123, b: 106, c: 100, d: 101 ✓, e: 240]

**Explanation:** cash realised = rs. ( 101.25 - 0.25 )✓ = rs

**Distribution** $p$ **from the PROXY LLM:** [0.1035, 0.0481, **0.6289**, 0.2061, 0.0122] → 'c' ✗

**Distribution** $\hat{p}$ **from the rectified flow model::** [0.2363, 0.0477, 0.0114, **0.6683**, 0.0567] → 'd' ✓

> An example on `MathQA` that a negative explanation generated by the PROXY LLM is optimized to a positive explanation after training for two rounds
>
> **Q** = A pet store regularly sells pet food at a discount of 10 percent to 30 percent from the manufacturer's suggested retail price. if during a sale, the store discounts an additional 20 percent from the discount price, what would be the lowest possible price of a container of pet food that had a manufacturer's suggested retail price of $ 40.00?
>
> **Decision set** $\mathcal{A}$ = [a: $ 10.00, b: $ 11.20, c: $ 22.40 ✓, d: $ 16.00, e: $ 18.00.]
>
> **Explanation:** retail price = manufacturer suggested retail price = $ 40 first discount price = retail price -10 % to -30 % = presence of retail price retrievable. ✗
>
> **Improved Explanation (Ours):** retail price =40 first maximum usually discounted price =40 - 30 % of 40 =40 - 12 =28 price after additional discounting during sale =28 - 20 % of 28 =28 - 5.60 =22.40 ✓

## I  LIMITATION

While the flow model is theoretically grounded and performs well in capturing reward distributions aligned with human preferences, it may still face generalization challenges in edge cases—such as ambiguous preference data, rare or unseen scenarios, and sensitive topics—similar to other LLM-based reward models.

Besides, our method assumes that decisions are context-dependent. For random actions, which lack a consistent rationale, the model cannot generate faithful explanations.

Moreover, learning from human feedback and ethical reasoning are two examples of such tasks that involve multiple plausible choices. Human feedback and ethical judgments may vary with different individuals, and the viewpoints of a population together form a complex distribution. Our method is suitable for such learning tasks. By contrast, our method may be less effective in domains such as abstract knowledge, where producing logical or factual rules is required. In such cases, constructing meaningful rewards from PROXY LLM s becomes more challenging.

Although our design choice foregoes direct analysis of internal mechanisms, we think that this pragmatic trade-off is warranted when developing methods intended to generalize across model classes with fundamentally different or opaque internals. It affords two advantages that are central to our goals: (i) it enables a unified treatment of diverse architectures—including RL policies and LLM agents—whose internal structures may be opaque, inaccessible, or highly heterogeneous; and (ii) it guarantees behavioral (functional) faithfulness, which is often a primary concern in safety-critical or deployment settings.

## J  IMPACT STATEMENTS

This paper presents work whose goal is to advance the field of machine learning by developing a model-agnostic explanation generator for intelligent agents, enhancing transparency and interpretability in agent decision prediction. The ability to generate effective and interpretable explanations has the potential to foster trust in AI systems, improving effectiveness in high-stakes applications such as healthcare, finance, and autonomous systems. Overall, we believe our work contributes positively to the broader AI ecosystem by promoting more explainable and trustworthy AI.

