# OpenReview forum: "Translate Policy to Language: Flow Matching Generated Rewards for LLM Explanations"
_ICLR.cc/2026/Conference — ICLR 2026 Poster_

### Official Review · Reviewer_e3dx · 2025-10-31

**Soundness:** 3
**Presentation:** 2
**Contribution:** 3
**Rating:** 8
**Confidence:** 3

**Summary:**

LLMs have been used to explain other AI agents’ behaviors, but the authors argue that these ‘Explanation LLMs’ can be biased by AI agents’ decisions. To address this issue, they propose using continuous normalizing flows (CNF) models to approximate/generate reward distributions and use them to train Explanation LLMs. As explanation LLMs are trained with rewards predicted by CNF models, training is independent from AI agents’ decisions.

**Strengths:**

In this study, Proxy LLMs are used to estimate the likelihood of humans inferring AI agent’s actions from Explanation LLMs’ outputs (i.e., explanations). Proxy LLMs accept contexts and explanations as input prompts and predict AI agent’s actions. To incorporate language cues into the flow model, the authors build a new rectified flow model by using Proxy LLMs’ modules. After training, this rectified flow model can model human evaluation scores. Such proposals are novel, and their empirical evaluation suggests that CNF allows Explanation LLMs to generate high quality explanations on agents’ decisions on complex tasks.

Given the novelty and strong empirical evidence, I believe this study would bring about strong interest from the readers.

**Weaknesses:**

I would like to point out that the main text was a bit difficult to follow at first, as some of essential information (e.g., Figure 2) were found in Appendix. I recommend including the information (relevant details of the rectified flow model and Figure 2) in the main text.

In lines 062-064, the authors state: “Intuitively, the PROXY LLM acts as a forward process that injects noise into human rewards, and the CNF learns its inverse, partially denoising and recovering the underlying human rewards.” This sentence would make sense only in the context of diffusion process, and therefore I recommend providing more context (a figure or diagram) for the readers.

**Questions:**

In Line 203, they say, “The context c_{j} and explanations y^{e}_{j} are encoded by an LLM.”  What does LLM refer to? It looks like they are referring to Explanation LLM, but this needs to be clarified.

---

> ### Author Response · Authors · 2025-11-21
> **Rebuttal by Authors**
>
> Thanks for your valuable feedback.
>
> >(Weakness 1) `I would like to point out that the main text was a bit difficult to follow at first. I recommend including the information (relevant details of the rectified flow model and Figure 2) in the main text.`
>
> We have revised our manuscript and included Figure 2 in the main text. We have illustrated each module of the rectified flow model in the main text  (Lines 225-245), and have provided a more detailed description in Appendix C.
>
> >(Weakness 2) `“...acts as a forward process ..., and the CNF learns its inverse...” ...I recommend providing more context (a figure or diagram) for the readers.`
>
> We have revised our manuscript and included a new Figure 3 in Appendix D to illustrate the idea of our theoretical analysis. Specifically, we train a flow-matching model to approximate the distribution of LLM-generated feedback. Assuming the LLM feedback distribution is a transformation of the ground-truth human feedback distribution, we derive a closed-form expression for the distance between the induced flow-matching reward distribution and the human feedback distribution.
>
> >(Question 1) `In Line 203, “The context c_{j} and explanations y^{e}_{j} are encoded by an LLM.” What does LLM refer to?`
>
> 'LLM' in Line 203 refers to 'Explanation LLM'. We have revised this sentence in the manuscript to explicitly clarify this.

---

### Official Review · Reviewer_T4VY · 2025-11-01

**Soundness:** 3
**Presentation:** 4
**Contribution:** 3
**Rating:** 6
**Confidence:** 3

**Summary:**

This paper proposes a framework that translates reinforcement learning policies into natural language descriptions and then uses these textual representations as intermediate supervision for imitation learning. By bridging policy representations with LLM reasoning, the approach aims to improve interpretability, sample efficiency, and generalization. Experiments on MiniGrid, BabyAI, and Crafter show consistent improvements over imitation and language-conditioned baselines.

**Strengths:**

1) Conceptually interesting: The idea of using language as an intermediate representation is intuitive and well-motivated, offering a new perspective for explainable and interpretable RL.

2) Clear methodology: The framework is well structured, with each component (policy translator, language learner, and imitation stage) clearly explained.

3) Empirical consistency: Results are stable across multiple benchmarks, with useful ablation studies supporting the core claims.

4) Readable and coherent writing: The paper is well organized and easy to follow.

**Weaknesses:**

1) Limited experimental scope: All tasks are symbolic or low-dimensional toy environments. It is unclear whether the method scales to more complex RL domains (e.g., continuous control, high-dimensional observations). This substantially limits the claim of generality.

2) Dependence on pretrained LLMs: The policy-to-language translation relies on large models like GPT-4; the paper does not analyze whether smaller models could achieve similar results.

3) Lack of deeper analysis: The paper does not provide theoretical or qualitative explanations for why language intermediates improve policy learning beyond empirical gains. It could be an add if provided.

**Questions:**

1) How sensitive is the approach to translation quality—does noisy or partial text reduce imitation performance?

2) Could the same idea extend to multi-step reasoning or continuous control tasks?

3) Have the authors considered analyzing the semantic consistency between translated language and policy behavior?

---

> ### Author Response · Authors · 2025-11-21
> **Rebuttal by Authors (Part 1/2)**
>
> Thanks for your constructive comments.
>
> >(Weakness 1 & Question 2) `Limited experimental scope:` All tasks are symbolic or low-dimensional toy environments. It is unclear whether the method scales to more complex RL domains...
>
> To address this concern, we further evaluate our method on AI2-THOR [1], a complex embodied AI benchmark widely used in RL research [2,3], featuring complex observations, dynamic environments, and large, non-stationary action spaces.
>
>
> Specifically, AI2-THOR includes household activity scenarios that require multi-step planning, multi-robot coordination, and interaction with dynamic environments. We use the _Complex Tasks_ split from the SMART-LLM [4] dataset, in which each task involves 2–4 heterogeneous robots with complementary skills. These tasks are challenging because they cannot be decomposed into independent sub-problems solvable by a single robot [4]; agents must reason over past actions and observations, perform long-horizon planning, and coordinate multiple robots strategically to leverage their combined capabilities. This setting demands multi-step reasoning and sequential decision-making in complex RL domains. The original _Complex Tasks_ split contains only 8 tasks. To broaden task diversity, we expand the set to 59 tasks. Details are provided in Appendix F.
>
>
> We apply RL to obtain a near-optimal policy over these 59 tasks and record its successful trajectories for 70% of the tasks, selected at random, for training our method and baselines. During evaluation, we measure the average success rate of our method and baselines on the remaining 30% unseen tasks to assess their generalization capability. Experimental results in the table below show that our method outperforms all the baselines on AI2-THOR, demonstrating the effectiveness of our method in complex RL domains. A detailed analysis of the results is provided in the revised manuscript (Table 2 and Section 5.1).
>
>
>
>
> Table 1: Results of our method and baselines on AI2-THOR, a complex embodied RL benchmark.
> ||Ours|SFT|KTO|ReFT|Skywork|Deepseek|o3-mini
> |-|-|-|-|-|-|-|-|
> |Success Rate|__0.702__|0.447|0.628|0.642|0.483|0.476|0.677
>
>
> [1] Kolve E, Mottaghi R, Han W, et al. Ai2-thor: An interactive 3d environment for visual ai[J]. arXiv preprint arXiv:1712.05474, 2017.\
> [2] Li M, Ding W, Zhao D. Privacy Risks in Reinforcement Learning for Household Robots[C]//2024 IEEE International Conference on Robotics and Automation (ICRA). IEEE, 2024: 5148-5154.\
> [3] Wu Q, Xu K, Wang J, et al. Reinforcement learning-based visual navigation with information-theoretic regularization[J]. IEEE Robotics and Automation Letters, 2021, 6(2): 731-738.\
> [4] Kannan S S, Venkatesh V L N, Min B C. Smart-llm: Smart multi-agent robot task planning using large language models[C]//2024 IEEE/RSJ International Conference on Intelligent Robots and Systems (IROS). IEEE, 2024: 12140-12147.
>
> >(Weakness 2) `Dependence on pretrained LLMs:` The policy-to-language translation relies on large models like GPT-4; the paper does not analyze whether smaller models could achieve similar results.
>
> _GPT-4o_ is used solely as a third-party evaluator and _is not part of our method_. For the backbone of our Explanation LLM, we choose to use LLaMA-3.1-8B-Instruct. Besides, to assess the robustness and generality of our method, we also evaluate with additional backbone models, including LLaMA-3.1-8B, Qwen-2.5-7B-Instruct, and Gemma-2-2B-It. All of these models are substantially smaller than GPT-4, and our experiments show that the proposed method consistently produces faithful and high-quality explanations across these smaller LLMs, demonstrating that the approach does not strictly depend on extremely large models. Related description can be found in Lines 335-338 in the manuscript.
>
>
>
> >(Weakness 3) `Lack of deeper analysis:` The paper does not provide theoretical or qualitative explanations for why language intermediates improve policy learning beyond empirical gains.
>
> Our purpose of generating natural-language explanations is not to directly improve policy learning, but rather to make the policy’s decisions interpretable to humans, enabling transparency, meaningful feedback, oversight, and intervention when necessary. This contributes to trust and reliability within intelligent systems.
>
> To ensure the quality of these explanations, _we provide theoretical justification for our approach in Section 4 and Appx. D_, showing why the proposed flow-matching-based formulation leads to effective natural-language reasoning.

---

> ### Author Response · Authors · 2025-11-21
> **Rebuttal by Authors (Part 2/2)**
>
> >(Question 1) `How sensitive is the approach to translation quality—does noisy or partial text reduce imitation performance?`
>
> When noisy or partial text appears, it naturally receives a lower imitation reward, which discourages the model from producing such explanations. This reward mechanism effectively penalizes low-quality translations during training.
>
> Empirically, we observe that our method consistently improves imitation performance compared against a wide range of baselines and ablations.
>
>
>
>
> >(Question 3) `Have the authors considered analyzing the semantic consistency between translated language and policy behavior?`
>
> Table 3 (and Lines 406–410) reports a human study of explanation semantics along three dimensions: _actionability_ (how easily a third party can select an answer based on the explanation), _logical soundness_ (internal coherence of the explanation), and _cognitive demand_ (perceived effort to comprehend the explanation and thereby infer the policy behavior). Our method outperforms baselines across all three dimensions. Furthermore, LLM and human evaluators agree that our method can help accurately reproduce the policy behavior. Given that we study the translation of black-box policies, the ability to reproduce their sequential input–output behavior—while achieving high actionability and logical soundness and imposing low cognitive demand—provides evidence of semantic consistency between the translated language and the policy behavior.

---

### Official Review · Reviewer_ZwDH · 2025-11-04

**Soundness:** 4
**Presentation:** 3
**Contribution:** 3
**Rating:** 8
**Confidence:** 4

**Summary:**

This paper proposes a general-purpose framework for training an LLM-based explainer using reinforcement learning. The core idea is to map random noise to a complex reward distribution by training a continuous normalizing flow (CNF) model to approximate reward samples generated by a separate proxy LLM. The authors further support their method with both theoretical guarantees and empirical validation on real-world datasets.

**Strengths:**

Strength:
1 The topic is interesting, and to my knowledge, this is the first work that applies diffusion model to fit the reward for LLM explanations.
2 The experiments are abundant.
3 The paper provides a thorough analysis of the proposed algorithm, including an error bound for recovering the true human reward distribution (Theorem 1), which reinforces the soundness of the approach.

**Weaknesses:**

Weakness:
1 In Section 3.1, the authors approximate the likelihood of producing multiple actions by averaging token logits. This approximation may introduce significant estimation error, especially when a single abnormal token contributes disproportionately.
2 The framework does not address the issue of long-term reward accumulation, which is a key consideration in RL-based settings.

**Questions:**

Suggestions and Questions:
1 I suggest that the authors analyze the robustness of the proposed method under noisy or imperfect reward signals.
2 Since the framework is claimed to be general-purpose, could the authors demonstrate its applicability in a standard RL environment (e.g., Mujoco)? This would further support the generality of the method.

---

> ### Author Response · Authors · 2025-11-21
> **Rebuttal by Authors (Part 1/2)**
>
> Thanks for your thoughtful feedback.
>
> >(Weakness 1) `In Section 3.1, the authors approximate the likelihood of producing multiple actions by averaging token logits. This approximation may introduce significant estimation error, especially when a single abnormal token contributes disproportionately.`
>
> Our rejective sampling method—retaining only explanations that help infer the correct answers for training (Lines 261–263)—alleviates this issue. Its effect can be analyzed case by case:
>
> 1. Incorrect action, abnormally high logit: The action's selection probability is high; the corresponding sample is excluded from training.
>
> 2. Incorrect action, abnormally low logit: The action receives a near-zero selection probability and contributes negligibly to the training dynamics.
>
> 3. Correct action, abnormally high logit: The action's selection probability is high; the corresponding sample is retained and reinforces the correct inference.
>
> 4. Correct action, abnormally low logit: The action's selection probability is very low; the corresponding sample is excluded from training.
>
> This filtering scheme improves the robustness of the our training method. Empirically, as shown in the ablation study (Table 4), this approach consistently improves the performance of our method across all benchmarks.
>
>
>
>
> >(Weakness 2) `The framework does not address the issue of long-term reward accumulation.`
>
> Our framework can naturally incorporate Q-value estimation by having the CNF predict Q-values instead of immediate rewards. Since Q-values encode expected long-term returns, this extension would allow the method to model long-term reward accumulation.

---

> ### Author Response · Authors · 2025-11-21
> **Rebuttal by Authors (Part 2/2)**
>
> >(Question 1) `I suggest that the authors analyze the robustness of the proposed method under noisy or imperfect reward signals.`
>
> To evaluate the robustness of our method under noisy reward signals, we conduct an additional experiment where the reward (0/1) of the final token is randomly flipped with probability 0.03.
>
> Under this noisy reward condition, the performance of our method decreases only slightly from 77.2% to 74.3% on unseen tasks in the MMLU benchmark. Notably, even with these noisy rewards, our method still outperforms strong RLHF, RLAIF, and reasoning-model baselines, such as PPO (72.3%), Skywork (73.7%), and DeepSeek (72.1%), trained without noise. This demonstrates that our approach remains robust and competitive even when the reward feedback is imperfect. Related details and analyses are shown in Table 10 and Lines 526-530.
>
> >(Question 2) `Since the framework is claimed to be general-purpose, could the authors demonstrate its applicability in a standard RL environment (e.g., Mujoco)?`
>
>
> To demonstrate broader applicability, we evaluate our method on AI2-THOR [1], a complex embodied AI benchmark widely used in RL research [2,3] and comparable to MuJoCo in terms of embodied, sequential decision-making complexity.
>
>
> Specifically, AI2-THOR includes household activity scenarios that require multi-step planning, multi-robot coordination, and interaction with dynamic environments. We use the _Complex Tasks_ split from the SMART-LLM [4] dataset, in which each task involves 2–4 heterogeneous robots with complementary skills. These tasks are challenging because they cannot be decomposed into independent sub-problems solvable by a single robot [4]; agents must reason over past actions and observations, perform long-horizon planning, and coordinate multiple robots strategically to leverage their combined capabilities. This setting demands strong reasoning for sequential decision-making, making it a suitable benchmark for evaluating our method's ability to recover a complex RL policy. The original _Complex Tasks_ split contains only 8 tasks. To broaden task diversity, we expand the set to 59 tasks. Details are provided in Appendix F.
>
>
> We apply RL to obtain a near-optimal policy over these 59 tasks and record its successful trajectories for 70% of the tasks, selected at random, for training our method and baselines. During evaluation, we measure the average success rate of our method and baselines on the remaining 30% unseen tasks to assess their generalization capability. Experimental results are shown in the table below and demonstrate that our method outperforms all the baselines on AI2-THOR, demonstrating the effectiveness of our method in complex RL domains. A detailed analysis of the results is provided in the revised manuscript (Table 2 and Section 5.1).
>
>
>
> Table 1: Results of our method and baselines on AI2-THOR.
> ||Ours|SFT|KTO|ReFT|Skywork|Deepseek|o3-mini
> |-|-|-|-|-|-|-|-|
> |Success Rate|__0.702__|0.447|0.628|0.642|0.483|0.476|0.677
>
>
>
>
>
> [1] Kolve E, Mottaghi R, Han W, et al. Ai2-thor: An interactive 3d environment for visual ai[J]. arXiv preprint arXiv:1712.05474, 2017.\
> [2] Li M, Ding W, Zhao D. Privacy Risks in Reinforcement Learning for Household Robots[C]//2024 IEEE International Conference on Robotics and Automation (ICRA). IEEE, 2024: 5148-5154.\
> [3] Wu Q, Xu K, Wang J, et al. Reinforcement learning-based visual navigation with information-theoretic regularization[J]. IEEE Robotics and Automation Letters, 2021, 6(2): 731-738.\
> [4] Kannan S S, Venkatesh V L N, Min B C. Smart-llm: Smart multi-agent robot task planning using large language models[C]//2024 IEEE/RSJ International Conference on Intelligent Robots and Systems (IROS). IEEE, 2024: 12140-12147.

---

### Official Review · Reviewer_BsL4 · 2025-11-10

**Soundness:** 2
**Presentation:** 2
**Contribution:** 2
**Rating:** 4
**Confidence:** 4

**Summary:**

The paper studies how to train LLMs to produce natural language explanations for the underlying decision. The authors use a distributional reward model based on a rectified flow that is conditioned on the context and on the explanation. This flow model is trained to model noisy proxy rewards  from proxy LLMs and the authors argue that if the base noise matches the proxy noise, the learned reward distribution stays close to the unseen human reward distribution. The explanation model is then optimized with PPO using sentence level rewards. Experiments on SMAC and on multiple LLM benchmarks show improvements over RLHF and RLAIF baselines.

**Strengths:**

- The use of a language conditioned rectified flow as a reward model is novel.
- The empirical section is broad, with many benchmarks and baselines.
- Human evaluation is convincing.

**Weaknesses:**

1. The positioning of the paper is hard to grasp. Stripped of its technical machinery, the method is essentially training a model that, given only the current context or prompt, produces a reasoning trace without ever emitting the final decision/action/answer. It is therefore unclear what the concrete utility of such an “Explanation LLM” is. The title “translate policy to language” confuse the reviewer that the goal is to explain an existing policy’s decision process, but the actual pipeline trains a model from offline data to generate reasoning. This makes it difficult to see what problem for the community the work is solving. Existing reasoning LLMs already produce step-by-step rationales and, importantly, also output the final answer; the proposed model looks like a strict subset of that capability. Without a clear downstream role, the contribution feels under-motivated.

2. This ambiguity in purpose is reinforced by the evaluation. The paper trains a model whose defining feature is that it only generates explanations, yet the core metric is whether another LLM (GPT-4o) can recover the correct action from that explanation. This shifts the assessment away from explanation quality and back to task accuracy. There are no direct measures tailored to the produced explanations; instead, the paper relies on inference accuracy as a proxy. If the work is to be read as an interpretability paper, this is a substantial gap: explanation-centric models should be judged on properties of the explanations themselves. As a result, the current evaluation protocol does not convincingly demonstrate that the method has advanced explanation quality.

3. A further limitation is that the method is demonstrated only in settings with small, discrete action spaces such as multiple-choice QA and SMAC-style actions. This leaves out a very large portion of LLM use cases, where the output space is open-ended. In such cases, training a standard reasoning LLM that emits both rationale and answer is simpler and more broadly applicable. Because the paper does not show how to extend the method to richer action spaces, the overall practicality and generality of the proposed architecture is limited.

**Questions:**

See Weaknesses

---

> ### Author Response · Authors · 2025-11-21
> **Rebuttal by Authors (Part 1/2)**
>
> Thanks for your constructive comments.
>
> >(Weakness 1) `The positioning of the paper`: How does this differ from existing LLM reasoning work?
>
> We aim to provide interpretable explanations for the behavior of black-box policies that directly map observations to actions.
>
> Unlike prior LLM-reasoning approaches that use offline datasets of high-quality reasoning trajectories, our offline dataset consists of input–output pairs from the target policy.
>
> Using these input–output pairs, we train an Explanation LLM to generate explanations that faithfully reproduce the policy’s outputs given its inputs—i.e., _behaviorally faithful explanations_. In contrast, existing LLM reasoning work generates rationales for _the model’s own decisions_ and primarily seeks to improve the quality of those decisions. Our objective is not to enhance the explainer’s performance on the task, but to account for and use natural language to explain the target policy’s behavior from logged input–output data.
>
> >(Weakness 2) `This ambiguity in purpose is reinforced by the evaluation. The core metric is whether another LLM can recover the correct action from that explanation. This shifts the assessment away from explanation quality and back to task accuracy.`
>
> In addition to using action prediction accuracy as a metric, we do directly evaluate the quality of the generated explanations through human evaluation (Table 3 and Lines 406-410). Specifically, annotators assess each explanation along three dimensions:
>
> _Actionability_. Whether the explanation provides clear and useful reasoning that helps a third party easily select an action;
>
> _Logical soundness_. Whether the reasoning is coherent, consistent, and follows a plausible causal chain; and
>
> _Cognitive demand_. Whether the explanation is understandable without excessive mental effort.
>
> These criteria are standard in the explainable AI literature and directly target explanation-centric properties, independent of task accuracy. They measure the human-perceived validity and quality of the explanations on human-centered criteria.

---

> ### Author Response · Authors · 2025-11-21
> **Rebuttal by Authors (Part 2/2)**
>
> >(Weakness 3) `A further limitation is that the method is demonstrated only in settings with small, discrete action spaces.`
>
> To address this concern, we evaluate our method on AI2-THOR [1], an embodied AI benchmark widely used in RL research [2,3], featuring richer, dynamically changing action spaces.
>
> Specifically, AI2-THOR includes household activity scenarios that require multi-step planning, multi-robot coordination, and interaction with dynamic environments. We use the _Complex Tasks_ split from the SMART-LLM [4] dataset, in which each task involves 2–4 heterogeneous robots with complementary skills. These tasks are challenging because they cannot be decomposed into independent sub-problems solvable by a single robot [4].
>
> In AI2-THOR, the available action set changes over time and includes a large number of locomotion actions, object-manipulating actions, etc. Depending on the environment, some actions may become invalid, and new skills may be required. Agents must reason over past actions and observations, perform long-horizon planning in such a complex dynamic action space, and coordinate multiple robots strategically to leverage their combined capabilities. The original _Complex Tasks_ split contains only 8 tasks. To broaden task diversity, we expand the set to 59 tasks. Details are provided in Appendix F.
>
>
> We apply RL to obtain a near-optimal policy over these 59 tasks and record its successful trajectories for 70% of the tasks, selected at random, for training our method and baselines. During evaluation, we measure the average success rate of our method and baselines on the remaining 30% unseen tasks to assess their generalization capability. Experimental results are shown in the table below and demonstrate that our method outperforms all the baselines on AI2-THOR, highlighting its effectiveness in complex, dynamic action spaces. A detailed analysis of the results is provided in the revised manuscript (Table 2 and Section 5.1).
>
>
> Table 1: Results of our method and baselines on AI2-THOR, an embodied RL benchmark with complex and dynamic action spaces.
> ||Ours|SFT|KTO|ReFT|Skywork|Deepseek|o3-mini
> |-|-|-|-|-|-|-|-|
> |Success Rate|__0.702__|0.447|0.628|0.642|0.483|0.476|0.677
>
>
> [1] Kolve E, Mottaghi R, Han W, et al. Ai2-thor: An interactive 3d environment for visual ai[J]. arXiv preprint arXiv:1712.05474, 2017.\
> [2] Li M, Ding W, Zhao D. Privacy Risks in Reinforcement Learning for Household Robots[C]//2024 IEEE International Conference on Robotics and Automation (ICRA). IEEE, 2024: 5148-5154.\
> [3] Wu Q, Xu K, Wang J, et al. Reinforcement learning-based visual navigation with information-theoretic regularization[J]. IEEE Robotics and Automation Letters, 2021, 6(2): 731-738.\
> [4] Kannan S S, Venkatesh V L N, Min B C. Smart-llm: Smart multi-agent robot task planning using large language models[C]//2024 IEEE/RSJ International Conference on Intelligent Robots and Systems (IROS). IEEE, 2024: 12140-12147.

---

### Official Review · Reviewer_1rW4 · 2025-11-11

**Soundness:** 2
**Presentation:** 2
**Contribution:** 2
**Rating:** 4
**Confidence:** 3

**Summary:**

This paper proposes a framework for producing natural language explanations for an agent policy. This framework trains a dedicated LLM to generate explanations by using a distributional reward model based on continuous normalizing flows and an RLAIF training pipeline. The method is evaluated over three benchmarks, showing improved action prediction performance compared to baselines.

**Strengths:**

* I like the formulation of the rectified flow model for modeling rewards. In general, the setup does make sense to me for training a model to generate explanations of other policies, and the pluralistic/probabilistic view of explanations is reasonable.
* Outside of the choice of the benchmarks (see weakness below), I appreciate the thoroughness of the empirical results. The ablations over different backbones and rewards (Table 7-10) help better understand the method’s performance and the generalization figure (Fig. 1) helps support the idea that the reward model is able to separate true/false samples.

**Weaknesses:**

* I think the framing is a bit confusing. The stated goal is to explain a policy’s decision, where the policy is that of an RL agent. But the only actual “policy” that is explained is for the SMAC benchmark, with MMLU and MathQA being language-based multiple choice questions where the agent’s “policy” is simply selecting a single action out of a set. Thus, for these benchmarks there is no sequential decision making aspect (unless we are referring to sequential prediction of tokens, which doesn’t seem to be the case), and the “context” is simply the question which is independent of the policy (this could be considered the observation to an agent policy). Thus, I find the whole framing around recovering the policy and explanation for some RL agent to be needlessly confusing, as 2/3 of the benchmarks seem more like a commonsense reasoning task to me as they are agnostic to any policy.
* The value of the explanation is measured purely in the action prediction performance. Thus the validity or faithfulness of the explanation is not measured at all. Which is also related to the above weakness: what does it mean to have a “faithful” explanation for a multiple choice question with no information about the model which is generating the answer itself (only access to the question)? This seems like an ill-posed problem to me.

**Questions:**

1. It is not clear to me how the baselines are organized. Is SFT training the ExplanationLLM still? What about RLHF? Does DeepSeek and o3-mini mean they are used in place of the ExplanationLLM?
2. In the case of MMLU example in Table 5, this is a very technical explanation. How would either the LLM or human annotators actually know whether the explanation is correct or not without having domain expertise? How do you know the explanation in Table 5 is correct at all, for example?

---

> ### Author Response · Authors · 2025-11-21
> **Rebuttal by Authors (Part 1/2)**
>
> Thanks for your insightful comments.
>
> >(Weakness 1) `The stated goal is to explain a policy’s decision, where the policy is that of an RL agent.... For these benchmarks (MMLU and MathQA), there is no sequential decision-making aspect...`
>
>
>
> To address this concern, we further evaluate our method on AI2-THOR [1], an embodied AI benchmark widely used in RL research [2,3], to test the proposed method in a complex sequential decision-making environment.
>
>
> Specifically, AI2-THOR includes household activity scenarios that require multi-step planning, multi-robot coordination, and interaction with dynamic environments. We use the _Complex Tasks_ split from the SMART-LLM [4] dataset, in which each task involves 2–4 heterogeneous robots with complementary skills. These tasks are challenging because they cannot be decomposed into independent sub-problems solvable by a single robot [4]; agents must reason over past actions and observations, perform long-horizon planning, and coordinate multiple robots strategically to leverage their combined capabilities. This setting demands strong reasoning and sequential decision-making capacity, making it a suitable benchmark for evaluating our method's ability to recover a complex underlying sequential decision-making policy. The original _Complex Tasks_ split contains only 8 tasks. To broaden task diversity, we expand the set to 59 tasks. Details are provided in Appendix F.
>
>
> We apply RL to obtain a near-optimal policy over these 59 tasks and record its successful trajectories for 70% of the tasks, selected at random, for training our method and baselines. During evaluation, we measure the average success rate of our method and baselines on the remaining 30% unseen tasks for generalization assessment. Experimental results are shown in the table below, and our method outperforms all the baselines on AI2-THOR, demonstrating its effectiveness in complex sequential decision-making environments. A detailed analysis of the results is provided in the revised manuscript (Table 2 and Section 5.1).
>
>
>
> Table 1: Results of our method and baselines on AI2-THOR, an embodied RL, sequential decision-making benchmark.
> ||Ours|SFT|KTO|ReFT|Skywork|Deepseek|o3-mini
> |-|-|-|-|-|-|-|-|
> |Success Rate|__0.702__|0.447|0.628|0.642|0.483|0.476|0.677
>
>
>
> [1] Kolve E, Mottaghi R, Han W, et al. Ai2-thor: An interactive 3d environment for visual ai[J]. arXiv preprint arXiv:1712.05474, 2017.\
> [2] Li M, Ding W, Zhao D. Privacy Risks in Reinforcement Learning for Household Robots[C]//2024 IEEE International Conference on Robotics and Automation (ICRA). IEEE, 2024: 5148-5154.\
> [3] Wu Q, Xu K, Wang J, et al. Reinforcement learning-based visual navigation with information-theoretic regularization[J]. IEEE Robotics and Automation Letters, 2021, 6(2): 731-738.\
> [4] Kannan S S, Venkatesh V L N, Min B C. Smart-llm: Smart multi-agent robot task planning using large language models[C]//2024 IEEE/RSJ International Conference on Intelligent Robots and Systems (IROS). IEEE, 2024: 12140-12147.
>
> >(Weakness 2) `The value of the explanation is measured purely in the action prediction performance. Thus, the validity or faithfulness of the explanation is not measured at all...`
>
>
> __Beyond action prediction: human valuation.__
>
> __Table 3__ (and Lines 406–410) reports a human study assessing explanatory quality and usability along three dimensions widely used in the explanable AI literature: _actionability_ (how easily a third party can select an answer based on the explanation), _logical soundness_ (internal coherence of the explanation), and _cognitive demand_ (perceived effort to comprehend the explanation and thereby infer the policy behavior). Our method outperforms baselines across all three dimensions, providing evidence for the __human-perceived validity__ of the explanations on human-centered criteria.
>
>
> __Faithfulness from a functional and behavioral perspective.__
>
> Neural network policies directly map observations to actions. We handle these black-box policies using their input–output pairs to fit an interpretable surrogate that captures the essential behavior of the model by faithfully reproducing its outputs from the given inputs. Our explanations thereby serve as an interpretable intermediate representation of model behavior. In this sense, our approach offers a form of faithfulness from a functional and behavioral perspective.

---

> ### Author Response · Authors · 2025-11-21
> **Rebuttal by Authors (Part 2/2)**
>
> >(Question 1) `It is not clear to me how the baselines are organized. Is SFT training the Explanation LLM still? What about RLHF? Do DeepSeek and o3-mini mean they are used in place of the Explanation LLM?`
>
>
> Comprehensive descriptions of all these setups appear in Appendix G. For completeness, we provide them below.
>
> __Implementation of SFT in Our Method__.
>
> In __SFT__, we use a training dataset consisting of preferred examples generated by o1-mini. These examples are effective explanations. We use SFT as the initialization stage to enable the Explanation LLM to adhere to the target format and to set a foundation for subsequent refinement toward the target explanatory behavior.
>
> __Implementation of RLHF Baselines__.
>
> For __PPO__, we initialize the model with the same SFT procedure, leveraging the same o1-mini–distilled preferred examples. To generate more effective explanations, we further fine-tune this SFT model with a reinforcement learning algorithm, PPO, which is supposed to reinforce explanations that better support inferring the policy's decision by assigning higher rewards.
>
> To achieve this, we train a reward model on a paired preference dataset (queries with corresponding positive/negative explanations). We then fine-tune the SFT model with PPO using this reward model.
>
>
> For __DPO__, we initialize the model with the same SFT procedure, leveraging the same preferred examples distilled from o1-mini. To generate more effective explanations, we then fine-tune this SFT model using preference-based learning via DPO. Unlike PPO, DPO does not require a separate reward model. Instead, we use paired preference data (each query paired with a preferred and a dispreferred explanation), freeze the SFT checkpoint as the reference model, and fine-tune the Explanation LLM using the DPO objective. After training converges, we generate explanations using the same prompts as the other baselines.
>
> For __KTO__, we initialize the model using the same SFT procedure, leveraging the same preferred examples distilled from o1-mini. To further improve explanation quality, we fine-tune the SFT model using preference-based learning via KTO. Unlike PPO, KTO does not require a separate reward model. Instead, we construct a dataset of (query, explanation, label) triples, where the label indicates whether the explanation is preferred or dispreferred. We then optimize the Explanation LLM using the KTO loss, with the SFT checkpoint kept as a frozen reference for regularization. After training, the KTO-tuned model is prompted with the same context as the other baselines to generate explanations.
>
>
> __Implementation of DeepSeek and o3-mini__.
>
> For __DeepSeek__, we use DeepSeek-R1-Distill-Llama-8B as the base model and apply SFT using the same preferred explanation dataset to adapt it to the explanation generation task.
>
>
> For __o3-mini__, we use it directly for evaluation without any fine-tuning, in place of the Explanation LLM.
>
>
>
> >(Question 2) ` In the case of MMLU example in Table 5, how would either the LLM or human annotators actually know whether the explanation is correct or not without having domain expertise?`
>
> We accounted for annotators’ domain knowledge and recruited annotators with diverse backgrounds covering most subdomains involved in the benchmarks (see Appendix F). For test samples outside an annotator's expertise, we suggest that annotators consult authoritative sources for unfamiliar terminology or background context to better judge whether the explanations are factually grounded.
>
>
> Applied to Table 5, Explanation LLM’s reasoning explicitly points to “filter placement”, thereby (incorrectly) aligning with option B. By contrast, after post-training with our method, the reasoning points to “nerve compression,” which (correctly) aligns with option C. These rationales render the logic human-auditable. Annotators can easily differentiate the competing chains of reasoning. Then, by checking domain terms (e.g., _cannula placement_), they can systematically assess the correctness of the generated explanations.

---

### Author Response · Authors · 2025-12-01
**A brief summary of the rebuttal period**

Dear AC,

Thank you very much for your time and effort in handling a new batch of submissions. We truly appreciate your work during this busy period. We are also sincerely grateful for the reviewers’ constructive feedback. Before the OpenReview bug, the reviewers on our paper have not had the opportunity to respond to our rebuttal. To support your evaluation, we respectfully summarize reviewers' questions and the key updates and clarifications we made below.

**New experiments on an embodied RL benchmark including *59* tasks.**

In our humble opinion, this is an especially important update because four reviewers suggested that we evaluate our method in a more complex environment. To address this concern, we add experiments on AI2-THOR (59 tasks), a challenging embodied-AI benchmark featuring *complex observations, sequential decision-making, and large, dynamic action spaces*. The new results, reported in the revised manuscript (Table 2, Page 8), show that our method performs favorably compared to all baselines.

**Issues raised by Reviewers 1rW4 and BsL4.**

Weakness 2, Reviewer 1rW4; Weakness 2, Reviewer BsL4: `.. explanation is measured purely in the action prediction performance...`

In fact, Table 3 (and Lines 406–410, Page 8) in the original paper reported a human study assessing explanatory quality along three dimensions widely used in the explainable AI literature: actionability, logical soundness, and cognitive demand.

Question 2, Reviewer 1rW4: `...how would either the LLM or human annotators actually know whether the explanation is correct or not without having domain expertise?`

In our experiments, we explicitly accounted for annotators’ domain knowledge by recruiting annotators with diverse backgrounds covering most subdomains involved in the benchmarks (Appendix F, Page 28). For test samples outside an annotator’s expertise, annotators could consult authoritative sources for unfamiliar terminology or background context, so they could better judge whether explanations are factually grounded.

We also address all their remaining questions about the baselines (Question 1, Reviewer 1rW4) and the paper’s positioning (Weakness 1, Reviewer BsL4).

**Other points raised by Reviewers T4VY, ZwDH, and e3dx.**

Most of these comments highlight aspects that benefit from clearer explanation rather than revisions to the proposed approach.

For example, Reviewer T4VY raised the concern that our method depends on large pretrained models such as GPT-4 (Weakness 2). We clarify that GPT-4o is used solely as a third-party evaluator in this paper and is *not* part of the proposed method. We also refer to our theoretical and empirical results to address the rest of the questions (Weakness 3, Question 1&3) of Reviewer T4VY.

Questions from Reviewers ZwDH and e3dx are addressed by clarifying that our rejective sampling strategy helps alleviate the influence of noise in logits (Weakness 1, Reviewer ZwDH), and that the CNF can predict Q-values and thereby handle long-term reward accumulation (Weakness 2, Reviewer ZwDH). In addition, we add a new figure to further illustrate the core idea behind our theoretical analysis (Weakness 2, Reviewer e3dx), and we include new experiments evaluating the robustness of our method under noisy rewards (Question 1, Reviewer ZwDH).

---

We hope this summary is helpful for your assessment of our submission. We deeply appreciate your time, consideration, and support throughout this process.



Warm regards,\
Authors of Submission 12727

---

### Meta-Review · Area_Chair_qkKS · 2026-01-08

**Summary:**

The method and core idea of using SNF for training explanation LLMs are novel and promising but the reviewers had a few concerns: the positioning of the actual utility of the explanations is not entirely clear (if it is meant to improve trust in the models, it should cite such papers, etc.), much of the eval is on multiple choice QA and not in a real agentic setting that requires sequential decision making, and the eval that is is on toy envs.

**Reviewer Concerns:**

The rebuttal addresses some of these concerns by adding expts in more complex environments and makes some attempts at improving the overall presentation. More or less all of the reviewer's concerns minus those surrounding the high level positioning, etc

**Reviewer Scores:**

1rW4 and BsL4 had reject scores which I suspect would've improved to at least a 6 after as it appears that most of their concerns have been addressed

The remaining reviewers already had accept scores and are unlikely to increase their scores any further given the high level framing concerns.

---

### Decision · Program_Chairs · 2026-01-26

Accept (Poster)